# Intuitive movement-based prosthesis control enables arm amputees to reach naturally in virtual reality

Effie Segas[1], Sébastien Mick[1,2], Vincent Leconte[1], Océane Dubois[1,2], Rémi Klotz[3], Daniel Cattaert[1], Aymar de Rugy[1]*

[1]Univ. Bordeaux, CNRS, INCIA, UMR 5287, Bordeaux, France; [2]ISIR UMR 7222, Sorbonne Université, CNRS, Inserm, Paris, France; [3]CMPR Tour de Gassies, Bruges, France

*For correspondence:
aymar.derugy@u-bordeaux.fr

**Abstract** Impressive progress is being made in bionic limbs design and control. Yet, controlling the numerous joints of a prosthetic arm necessary to place the hand at a correct position and orientation to grasp objects remains challenging. Here, we designed an intuitive, movement-based prosthesis control that leverages natural arm coordination to predict distal joints missing in people with transhumeral limb loss based on proximal residual limb motion and knowledge of the movement goal. This control was validated on 29 participants, including seven with above-elbow limb loss, who picked and placed bottles in a wide range of locations in virtual reality, with median success rates over 99% and movement times identical to those of natural movements. This control also enabled 15 participants, including three with limb differences, to reach and grasp real objects with a robotic arm operated according to the same principle. Remarkably, this was achieved without any prior training, indicating that this control is intuitive and instantaneously usable. It could be used for phantom limb pain management in virtual reality, or to augment the reaching capabilities of invasive neural interfaces usually more focused on hand and grasp control.

## eLife assessment

This paper's **importance** lies in its integration of movement and contextual information to control a virtual arm for individuals with upper-limb differences. The provided evidence **convincingly** demonstrates the approach's feasibility for manipulating a single object shape in different orientations within a virtual environment. However, additional improvements are needed for this proof-of-concept neuro-model to fulfil practical requirements.

## Introduction

The field of bionic limbs has seen great progress over the last few years, including Targeted Muscle Reinnervation (TMR) (**Kuiken, 2009**), osseointegration (**Jönsson et al., 2011**), chronically implanted sensors and stimulators for bidirectional communication with the nervous system (**Ortiz-Catalan et al., 2020**; **Ortiz-Catalan et al., 2014a**; **D'Anna et al., 2019**; **Zollo et al., 2019**; **Salminger et al., 2019**), and advanced signal processing to encode and decode sensorimotor signals (**Jezernik et al., 2001**; **Farina et al., 2023**; **Cracchiolo et al., 2020**). Yet, simultaneous control of the multiple Degrees of Freedom (DoFs) of a prosthetic arm remains challenging, especially to bring a prosthetic hand to the correct location and orientation to efficiently grasp objects. Indeed, although simultaneous and proportional real-time myoelectric control has been achieved for two to three DoFs (**de Rugy et al., 2012**; **Jiang et al., 2012**; **Hahne et al., 2015**; **Ameri et al., 2014**; **Ortiz-Catalan et al., 2014b**; **Smith**

*et al., 2016*; *Nowak et al., 2022*), difficulties appear (*Hahne et al., 2015*) and performance deteriorates as the number of DoFs increases (e.g. success rate dropped from 96 to 37% from one to three DoFs in *Nowak et al., 2022*). Furthermore, this was achieved in lab settings, mostly on participants with valid arms, sometimes including few participants with limb difference at transradial level, either amputation or congenital limb difference (*Jiang et al., 2012*; *Hahne et al., 2015*; *Ameri et al., 2014*; *Hahne et al., 2018*), using native muscles remaining that normally actuate the DoFs under myoelectric control (forearm, wrist, hand).

In the case of an amputation at the humeral level, none of the forearm wrist, and hand muscles would remain, and the important additional DoF of the elbow would need to be controlled. Although TMR could be used to recover valid control signals by transferring residual arm nerves controlling missing distal muscles and joints to compartments of remaining muscles (*Kuiken, 2009*; *Kuiken et al., 2007*), this is usually geared toward recovering forearm supination-pronation and hand opening-closing (*Ortiz-Catalan et al., 2020*; *Hargrove et al., 2017*), arguably more important and available in commercial prosthesis, rather than wrist DoFs (flexion-extension and radial-ulnar deviation) which are nevertheless critical to orient the hand in space (*Montagnani et al., 2015a*; *Kanitz et al., 2018*). In the end, no myoelectric solution exists to simultaneously and intuitively control the four arm DoFs (from elbow to wrist included) that are necessary for people with transhumeral limb loss in order to correctly position and orient their prosthetic hand to grasp objects in a large reachable space.

Here, we provide a solution with an alternative movement-based approach which leverages natural coordination between arm segments and knowledge of the movement goal. Control strategies exploiting natural synergies in arm coordination (*Soechting and Lacquaniti, 1981*; *Desmurget et al., 1995*) have already been used to predict distal joints from the motion of proximal ones (*Popovic and Popovic, 2001*; *Kaliki et al., 2013*; *Merad et al., 2020*; *Merad et al., 2018*; *Montagnani et al., 2015b*). However, these have been mostly confined to the control of one DoF (i.e. either the elbow [*Popovic and Popovic, 2001*; *Merad et al., 2020*; *Merad et al., 2018*] or the wrist supination-pronation [*Montagnani et al., 2015b*], reconstructed from shoulder movement), or relying on additional unnatural movements to increase functionality (*Kaliki et al., 2013*). Here, we unleash this movement-based approach by adding knowledge of movement goals, which could be made available through computer vision (*Liu, 2022*; *Nguyen et al., 2022*) combined with gaze information (*Markovic et al., 2014*; *Pérez de San Roman et al., 2017*; *González-Díaz et al., 2019*). We showed recently that adding target position and orientation to an Artificial Neural Network (ANN) trained to predict four arm distal DoFs (elbow to wrist) from proximal (shoulder) motion greatly improves these predictions, as well as human-in-the-loop control using them (*Mick et al., 2021*). Yet, performance remained lower than natural movements, with increased compensatory movements from trunk and shoulder, a limited workspace, and a control design not directly applicable to people with transhumeral limb loss (*Mick et al., 2021*). Critical changes were brought about here to overcome all those limitations, and enabled 29 participants (including seven with transhumeral limb loss) to perform as well as natural, without any prior training, at picking and placing a bottle in a wide reachable space in virtual reality. A physical proof of concept is also provided whereby 15 participants, including three with limb deference, were able to reach and grasp real objects at different positions and orientations with a robotic arm operated according to the same control principle.

## Results

### Natural arm movement in virtual reality

After linking movement trackers placed on participants to that of a virtual arm, subjects were engaged in repeatedly picking a bottle standing on a platform and placing it on another platform (see *Figure 1b*, Methods, Virtual Arm Calibration, and Task). To maximize the workspace of this task, 300 'plausible' target locations (i.e. position and orientation) were determined randomly within the full range of motion established individually for each joint of each participant (see *Figure 1a*, Methods). Natural arm movements recorded to pick and place bottles at those 300 targets were then used to train the ANN involved in our movement-based control (*Figure 1c*, *cf* next section), but also to establish a new set of 200 'possible' targets within the workspace actually covered by the participants' arm during the initial acquisition (*Figure 1d*). Although 'plausible' targets are all supposed to be reachable, in practice, anatomical joints' limits are interdependent in a way that makes it uncomfortable or impossible

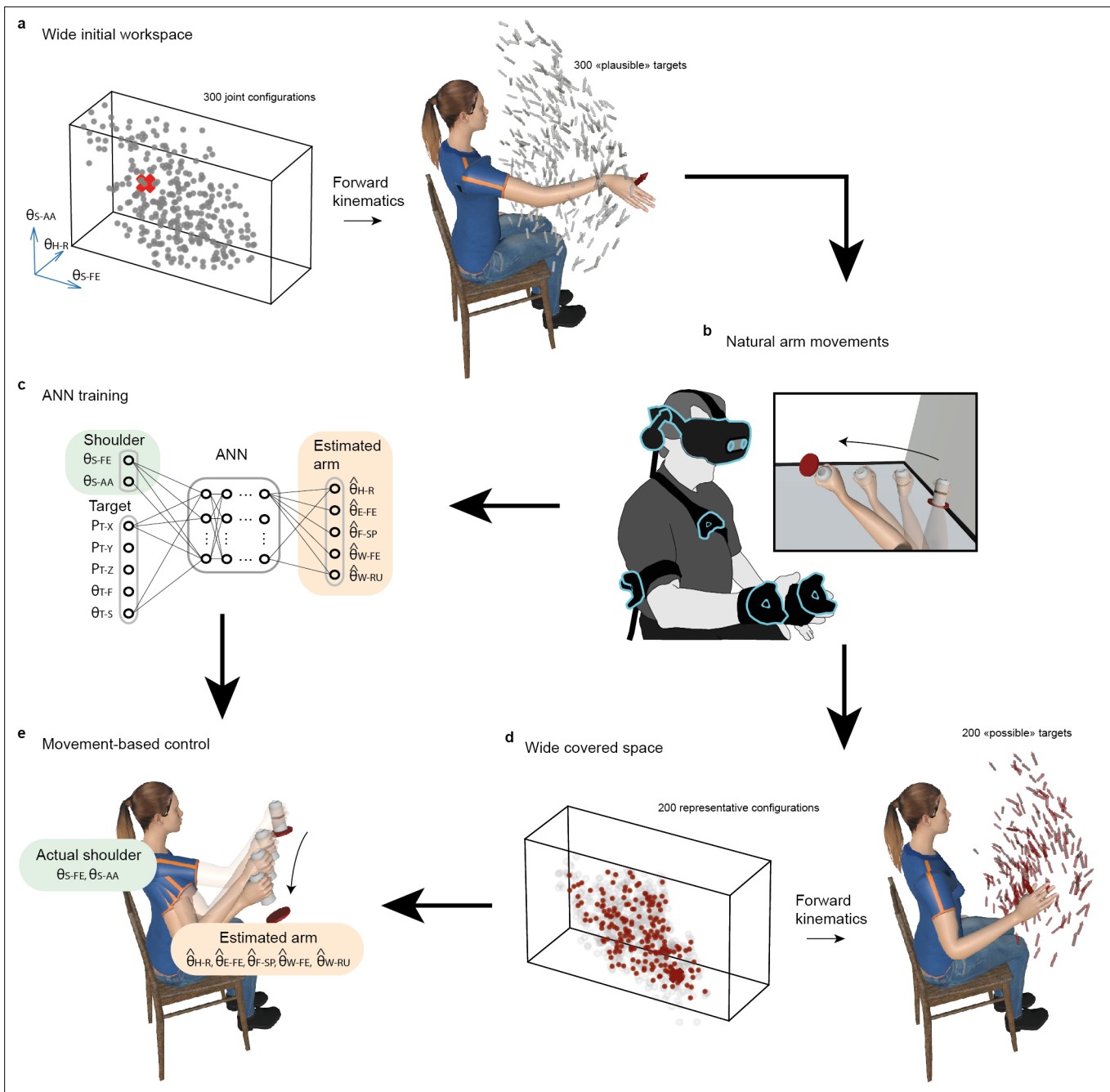

**Figure 1.** Overview of the task and control strategy. (**a**) Wide initial workspace. Three hundred 7-DoF arm configurations (gray dots, only three angles displayed for convenience) within the joint operating range of a given participant (materialized by the parallelepiped) are transformed into 300 plausible target locations (gray arrows) using forward kinematics. (**b**). Natural arm movements are recorded while participants equipped with movement trackers on the arm and torso are involved in picking and placing a bottle at the 300 target locations in virtual reality. (**c**). The Artificial Neural Network (ANN) is trained on recorded natural arm movement to reconstruct distal Degrees of Freedom (DoFs) (orange) from proximal ones (green) plus target information (position and orientation). (**d**). Wide space covered during recorded natural arm movements. Two hundred nodes (red dots) that best represent the arm angular configurations actually produced (gray circles) by a participant during her/his recorded natural arm movements were identified using an unsupervised self-organizing neural network, and transformed into a set of 200 possible targets (red arrows) using forward kinematics. (**e**). Movement-based prosthesis control. The participant performs the pick and place task at the 200 possible targets using a hybrid arm reproducing in real-time her/his own shoulder movements (green angles), and using the ANN predictions for the five remaining distal DoFs (orange angles). *Figure 1—figure supplement 1* provides complementary information about the ANN inputs and outputs, and the process used to remap data for different arm morphologies.

The online version of this article includes the following figure supplement(s) for figure 1:

**Figure supplement 1.** Overview of angular and movement-goal information.

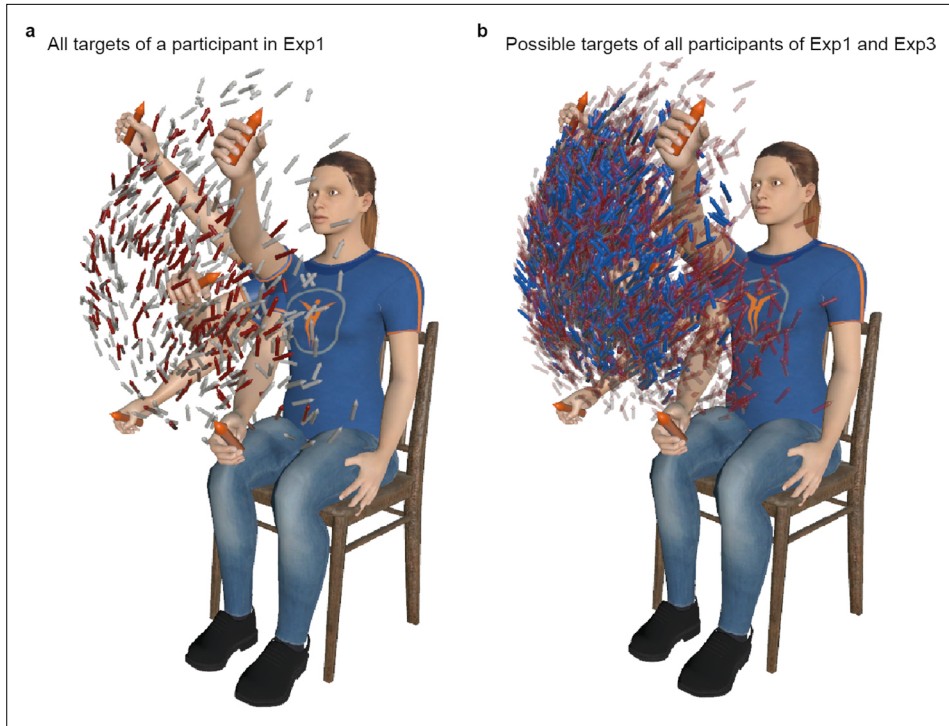

**Figure 2.** Wide workspace covered in experiments. (**a**). All targets used for a representative participant of Exp1 are displayed, together with five arm postures (four at extended positions and one flexed in the middle) to provide perspectives. Gray arrows represent plausible targets (n=300), and red arrows represent possible targets (n=200). (**b**). Possible targets of all participants of Exp1 (n=2000, red arrows) and Exp3 (n=1400, in blue arrows), remapped for an average arm, and regrouped on the same graph. Note that for Exp3, possible targets corresponding to participants with left-sided limb loss were mirrored to be represented in relation to a right arm. This figure illustrates the comparably large workspaces obtained for the 10 participants with intact limbs of Exp1 (used for the Generic ANN) and the seven participants with transhumeral limb loss of Exp3 (using the Generic Artificial Neural Network (ANN)).

to produce arm configurations with maximal excursion simultaneously at multiple joints. To circumvent this, we used an unsupervised self-organizing neural network (*Fritzke, 1994*) to identify 200 nodes that best represent the arm postures actually produced by participants, and used forward kinematics to turn these nodes into target locations, thereby obtaining a set of 200 possible targets guaranteed to be reachable with natural arm movements (*Figure 1d*, see Methods).

While 'plausible' targets were used in the initial acquisition to collect natural arm movements from which the movement-based prosthesis control was designed, 'possible' targets were used in all experimental test phases involved to compare the different controls. To illustrate the wide resulting workspace, *Figure 2a* shows the sets of plausible and possible targets of a representative participant, and *Figure 2b* the sets of possible targets for all subjects of Exp1 and Exp3. *Video 1* illustrates a participant completing the task at a comfortable yet sustained pace, representative of the overall performance observed in our experiments (i.e. with typical movement times of approximately 1.3 s between pick and place).

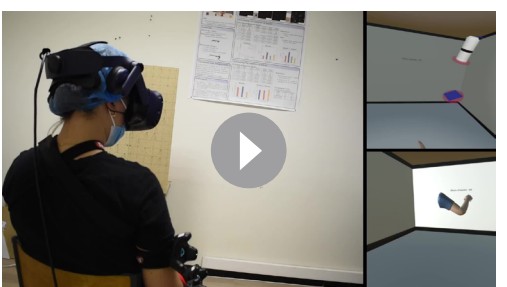

**Video 1.** A representative intact-limbs participant performed the pick and place task.
https://elifesciences.org/articles/87317/figures#video1

## Intuitive movement-based prosthesis control

From natural movements recorded in the initial acquisition, the control was developed based on an ANN trained to reconstruct distal joint angles

from shoulder kinematics plus contextual target information. An initial version of this control was proposed and tested in *Mick et al., 2021*, and critical changes were designed here to improve its quality and applicability to people with transhumeral limb loss. After being trained to reconstruct five distal arm angles from proximal shoulder kinematics plus target location (3D position and 2D orientation, see Methods), the ANN illustrated in *Figure 1c* was used to control a Hybrid Arm emulating the behavior of a residual upper arm fitted with a transhumeral prosthesis (*Figure 1e*). To this end, shoulder flexion-extension and abduction-adduction of the Hybrid Arm were operated from real

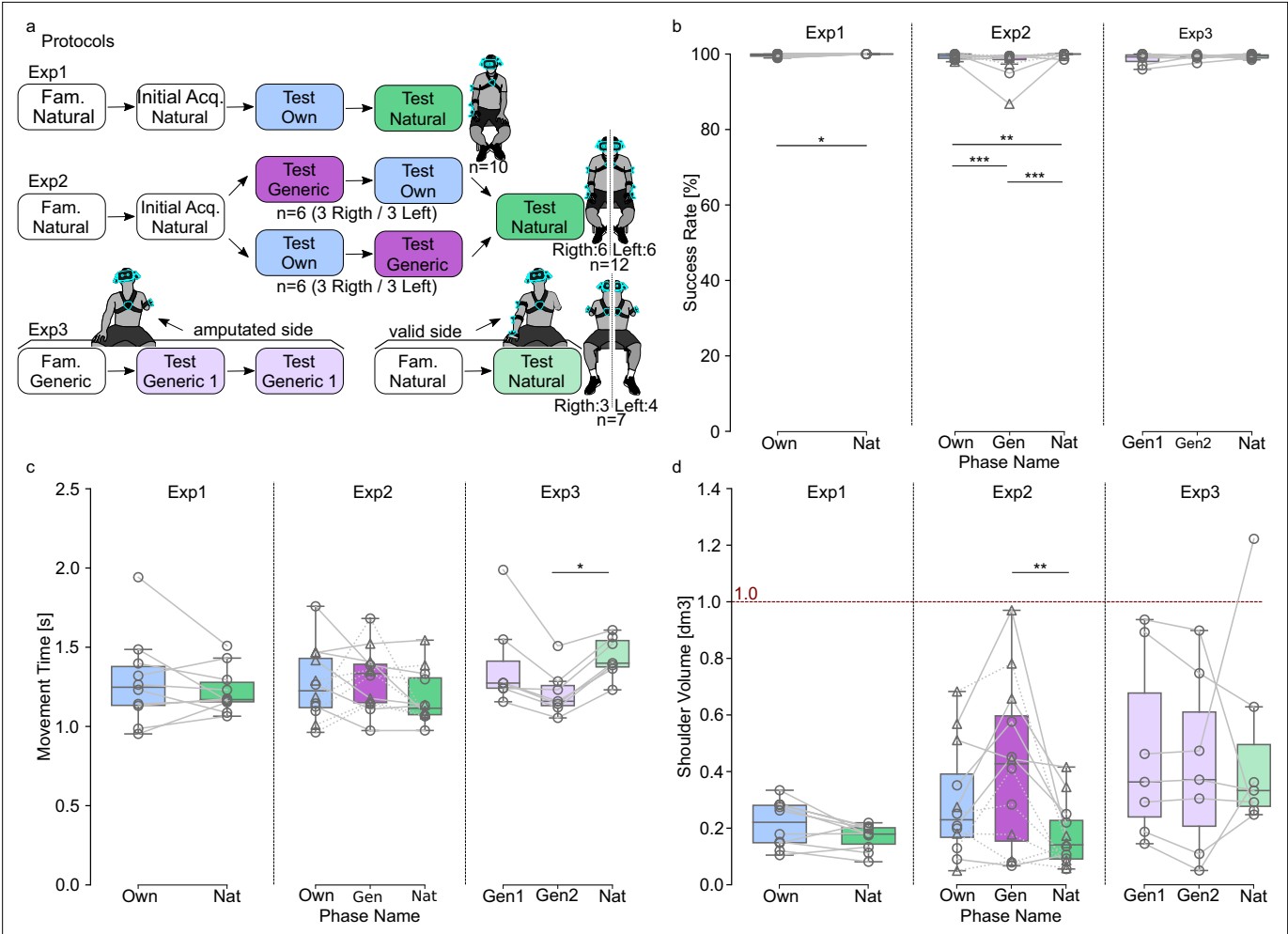

**Figure 3.** Protocols and results. (**a**) Protocols of the three experiments. Each box contains a phase name and the name of the control used. Fam. stand for Familiarization phase, and Initial Acq. for Initial Acquisition phase. The order of test phases conducted with the Own and the Generic Artificial Neural Networks (ANNs) was counterbalanced in Exp2. (**b–d**). Results for success rate (**b**), movement time (**c**) and shoulder volume (**d**). Each gray line corresponds to a participant. In Exp2, dashed lines indicate participants who began by the control with the Generic ANN, and plain lines those who began by the control with the Own ANN. Boxes limits show first and third quartiles whereas inside line shows the median value. Whiskers show min and max values. Own, Gen, Nat represent phases in which the control was performed with the Own ANN, the Generic ANN, and the Natural Virtual Arm, respectively. In Exp3, Gen1 and Gen2 refer to the first and second blocks performed with the Generic ANN. Stars represent significant differences, with * for p<0.05, ** for p<0.01, and *** for p<0.001. The dashed red line represents a volume of 1 dm3 (=1 L). *Figure 3—figure supplements 1–3* provide the participants' individual distributions of movement times for each of the three experiments, respectively.

The online version of this article includes the following figure supplement(s) for figure 3:

**Figure supplement 1.** Distributions of movement times of each participant (and data from all participants regrouped in the last subplot) for the two experimental conditions (TestNat and TestOwn) of Exp1.

**Figure supplement 2.** Distributions of movement times of each participant (and data from all participants regrouped in the last subplot) for the three experimental conditions (TestNat, TestOwn, and TestGen) of Exp2.

**Figure supplement 3.** Distributions of movement times of each participant (and data from all participants regrouped in the last subplot) for the three test phases (TestNat, TestGen1, and TestGen2) of Exp3.

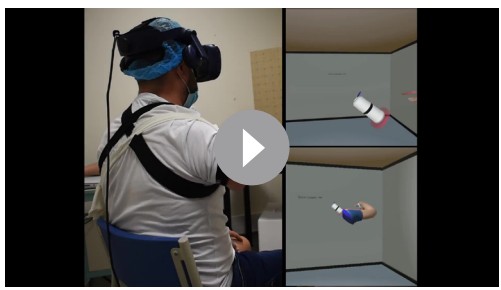

**Video 2.** Participant 1 with transhumeral limb loss performed the pick and place task with residual limb movement-based control. For anonymization purposes, the participant number indicated here is not the same as in *Table 1* nor as in the result section.
https://elifesciences.org/articles/87317/figures#video2

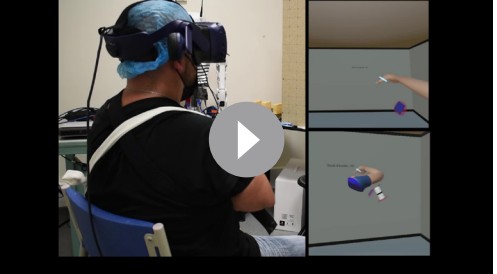

**Video 3.** Participant 2 with transhumeral limb loss performed the pick and place task with residual limb movement-based control. For anonymization purposes, the participant number indicated here is not the same as in *Table 1* nor as in the result section.
https://elifesciences.org/articles/87317/figures#video3

shoulder movements produced by the operator, whereas the five remaining joint angles were driven by predictions from the ANN. Importantly, the ANN could be trained either solely from the operator's own natural movements (Own ANN, see Methods) or from those of multiple other participants (Generic ANN). *Figure 3a* summarizes the protocols of the three experiments gradually leading to a functional control solution for prosthesis users. *Videos 2–4* illustrates the ultimate goal reached in this study, that is, to enable participants with transhumeral limb loss to pick and place objects with movement time and performance similar to that with a natural arm.

## Specific control based on participants' own movements

The first experiment aimed at (i) emulating a prosthesis control as intuitive as possible in a wide workspace for participants based on their own natural movements, and (ii) collecting natural movements from several subjects to build a generic model for prosthesis control to be tested on other participants with intact limbs or with limb difference (in Exp2 and 3, respectively). After an initial acquisition in which they picked-and-placed 300 plausible targets with their right arm in a wide workspace, 10 right-handed subjects were tested on 200 possible target locations with our intuitive prosthesis control trained on their own movements (Exp1, TestOwn, *Figure 3a*) before being tested again on the same targets with their natural arm movements (TestNat).

Results indicate high success rates for both conditions (median success rate of 100% and 99.7% for TestNat and TestOwn, respectively), with a minor (<1%) albeit significant difference between them (TestOwn vs TestNat; n=1886; McNemar test, p=0.023, df = 1; *Figure 3b*). Regarding movement times (i.e. time taken to reach and validate each target from the previous one), results were also closely similar and not significantly different between conditions (TestOwn vs TestNat; n=10; medians of 1.25 s vs 1.17 s, respectively; two-tailed paired t-test, p=0.378, t=–0.927, df = 9). Visual inspection of individual distributions of movement times provided *Figure 3—figure supplement 1* indicates that despite slight differences observed for some participants, no specific pattern emerges, and distributions look very similar between conditions when data from all participants are pooled together. The volume spread by the shoulder's trajectory throughout a phase, which includes compensatory movements of the body that might be elicited to compensate for imperfect control (see *Mick et al., 2021*), was also comparable and not significantly different between conditions (TestOwn vs TestNat; n=10; medians of 0.22 dm3 vs 0.18 dm3, respectively; two-tailed paired t-test, p=0.058, t=–2.168, df = 9).

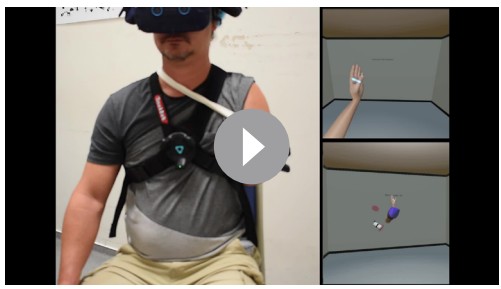

**Video 4.** Participant 3 with transhumeral limb loss performed the pick and place task with residual limb movement-based control. For anonymization purposes, the participant number indicated here is not the same as in *Table 1* nor as in the result section.
https://elifesciences.org/articles/87317/figures#video4

Overall, our movement-based control trained on the own natural movements of each participant enabled an almost perfect success rate (99.7%), and movement times similar to that with their natural arm to pick and place the bottle in the wide workspace tested. Yet, this control is inapplicable 'as is' to individuals with limb differences, for whom recording natural movements is obviously not possible from their missing arm. The next step was, therefore, to establish an equivalent control from data collected on the multiple participants of Exp1, and to be used on new naive participants in Exp2.

## Generic control based on movements from other participants

After building a generic model from data recorded in Exp1 (cf Methods), the second experiment aimed at (i) assessing performance with this generic model (as compared to that with natural movements or with a control based on each participant's own natural movements) and (ii) validating the use of this generic model on left-handed participants using their dominant arm. As the relationship between hand locations and arm configurations depends on segment dimensions that differ between participants, a critical step toward building an efficient generic model for a new user was to remap hand position data from previous participants to the arm dimensions of that new user (*Figure 1—figure supplement 1b*). This was done using forward kinematics to remap data from all participants of Exp1 before using them to train the generic ANN to be used for intuitive control in TestGeneric by each participant of Exp2 (see Methods). After an initial acquisition phase equivalent to that of Exp1, 12 participants (six left-handed) were tested on our movement-based control using either the generic model (TestGeneric) or the model based on their own movements (TestOwn). The order between TestGeneric and TestOwn was counterbalanced among subjects, with three left-handed participants in each group (see *Figure 3a*). The protocol ended with each participant being tested again with their natural arm movements (TestNat).

Participants achieved high success rates in all conditions (median success rates of 99% or higher, *Figure 3b*), although minor differences between conditions (<1%) were found significant (TestOwn vs TestGeneric vs TestNat; n=2268; medians of 99.50% vs 99.24% vs 100%, respectively, Cochran's Q test, $p=5.19.10^{-12}$, Q=51.97, df = 2; post-hoc McNemar test with Bonferroni adjustment; TestOwn vs TestGeneric, $p=6.42.10^{-5}$, chi.sq=18.1, df = 1; TestOwn vs TestNat, $p=8.58.10^{-3}$, chi.sq=8.89, df = 1; TestGeneric vs TestNat, $p=3.24.10^{-10}$, chi.sq=41.7, df = 1). Median movement times remained below 1.4 s in all conditions (*Figure 3c*), with no significant difference found between conditions (TestOwn vs TestGeneric vs TestNat; n=12; RM ANOVA test, p=0.181, DFn = 2, DFd = 22, *F*=1.848). Visual inspection of individual distributions of movement times provided *Figure 3—figure supplement 2* indicates that despite slight differences observed for some participants, no specific pattern emerges, and distributions look very similar between conditions when data from all participants are pooled together. With respect to the volume spread by the shoulder during the Test phases (*Figure 3d*), the statistical analysis revealed a significant effect of condition, (TestOwn vs TestGeneric vs TestNat; medians of 0.23dm3 vs 0.43 dm3 vs 0.14 dm3, respectively, n=12; RM ANOVA test, p=0.003, DFn = 2, DFd = 22, *F*=7.806), with post-hoc tests indicating a higher volume for TestGeneric than for Test Nat (Tukey test with Bonferroni correction; TestOwn vs TestNat, p=0.385; TestGeneric vs TestNat, p=0.025; TestOwn vs TestGeneric, p=0.347). Despite this difference, the spread volumes remained contained, never exceeding 1 liter (*Figure 3d*), which appears reasonable given the high compensations that could have occurred with less efficient controls over the wide workspace spanned by the targets.

Overall, our control based on a generic model trained on data from other participants enabled, therefore, new participants to reach almost all targets (>99%) as well as with their natural arm (or an intuitive control trained on their own natural movements) regardless of handedness, with a moderate increase in compensatory movements. The only remaining step was then to validate this control directly on individuals with limb differences.

## Successful validation on individuals with limb loss

As we now have an intuitive control applicable to participants with an arm amputation on either side of their body, we tested it on seven participants with unilateral transhumeral limb loss, three of them disabled on their left side. As no initial acquisition phase was conducted on these participants, the set of 200 possible targets was determined for each of them using a self-organizing map to extract representative postures from a large number of postures generated by movements produced by all participants of Exp1 filtered to the range of motion of their residual limb (see Methods). Given the

**Table 1.** Exp3 participants' amputation description.

Each line contains the time since amputation, the residual limb circumference and length, and the side of the amputation for a participant (*R*=right, L=left).

| Participants with transhumeral limb loss | Time since amputation (months) | Residual limb circumference (cm) | Residual limb length (cm) | Amputated arm side |
|---|---|---|---|---|
| Participant 1 | 20 | 33 | 15 | R |
| Participant 2 | 48 | 30 | 25 | L |
| Participant 3 | 12 | 30 | 35 | L |
| Participant 4 | 132 | 34 | 23 | R |
| Participant 5 | 120 | 35 | 30 | R |
| Participant 6 | 276 | 31 | 28 | L |
| Participant 7 | 108 | 32 | 23 | L |

lack of initial acquisition, participants had much less practice with the task before they were tested. Therefore, two blocks of intuitive control with the Generic ANN were conducted on their amputated side (TestGeneric 1 and 2, *Figure 3*), before they were tested on the same mirrored targets with their valid arm (TestNat).

As for other experiments, success rates were very high (all medians above 99%), and did not differ between conditions (TestGeneric1 vs TestGeneric2 vs TestNat; medians of 99.24% vs 99.50% vs 99.50%, respectively, n=1128; Cochran's Q test, p=0.1146, Q=4.33, df = 2). Movement times with our movement-based control were also in the same range as in previous experiments, and were even smaller by the second block of intuitive control (TestGeneric2) than with their valid arm (TestGeneric1 vs TestGeneric2 vs TestNat; medians of 1.27 s vs 1.16 s vs 1.40 s; n=7; Friedman test, p=0.002, chi-squared=12.286, df = 2; post-hoc Conover test; TestGeneric1 vs TestNat, p=0.619; TestGeneric2 vs TestNat, p=0.014; TestGeneric1 vs TestGeneric2, p=0.161). Visual inspection of individual distributions of movement times provided *Figure 3—figure supplement 3* confirms that despite the slight differences between participants, movement times tend to be smaller for TestGeneric phases (and even more so for TestGeneric2) than for TestNat. The volumes spread by the shoulder were comparable to that measured with the generic model on Exp2 (median 0.43 dm3) and did not differ between conditions (TestGeneric1 vs TestGeneric2 vs TestNat; medians of 0.36 dm3 vs 0.37 dm3 vs 0.33 dm3; n=7; Friedman test, p=0.867, chi-squared=0.28571, df = 2).

Overall, without any prior experience with the task nor with the apparatus, our intuitive control allowed disabled participants to achieve performance as good as with their valid arm, and to achieve comparable levels of performance as for other subjects in all conditions tested (see *Figure 3b–d*). Furthermore, beyond objective performance measures, our intuitive control elicited high enthusiasm from participants with limb loss. This is well illustrated by the following oral reports they formulated either spontaneously or during informal talks with the experimenter while the experiment was on pause or finished (for anonymization purposes the numbers reported are not the ones of *Table 1* or the *Videos 2–4*, and the number of years since amputation has been replaced by *X*):

Participant 1: 'A prosthesis that would be controlled like that? I take it right away.'

Participant2: 'At the end, it is quite intuitive.'

Participant 3 reported that he would use a prosthesis behaving like this.

Participant 4: 'Doing a real arm movement, this is enjoyable.' Participant 4 also reported that he found the arm a bit too stiff from time to time.

Participant 5: 'It's intuitive, it's easy.'

Participant 6: 'The movement doesn't feel like a natural movement to me, it's when I'm on target that the wrist is well placed.'

Participant 7: 'It's surprising. *X* years since I was able to do that, this is moving.'

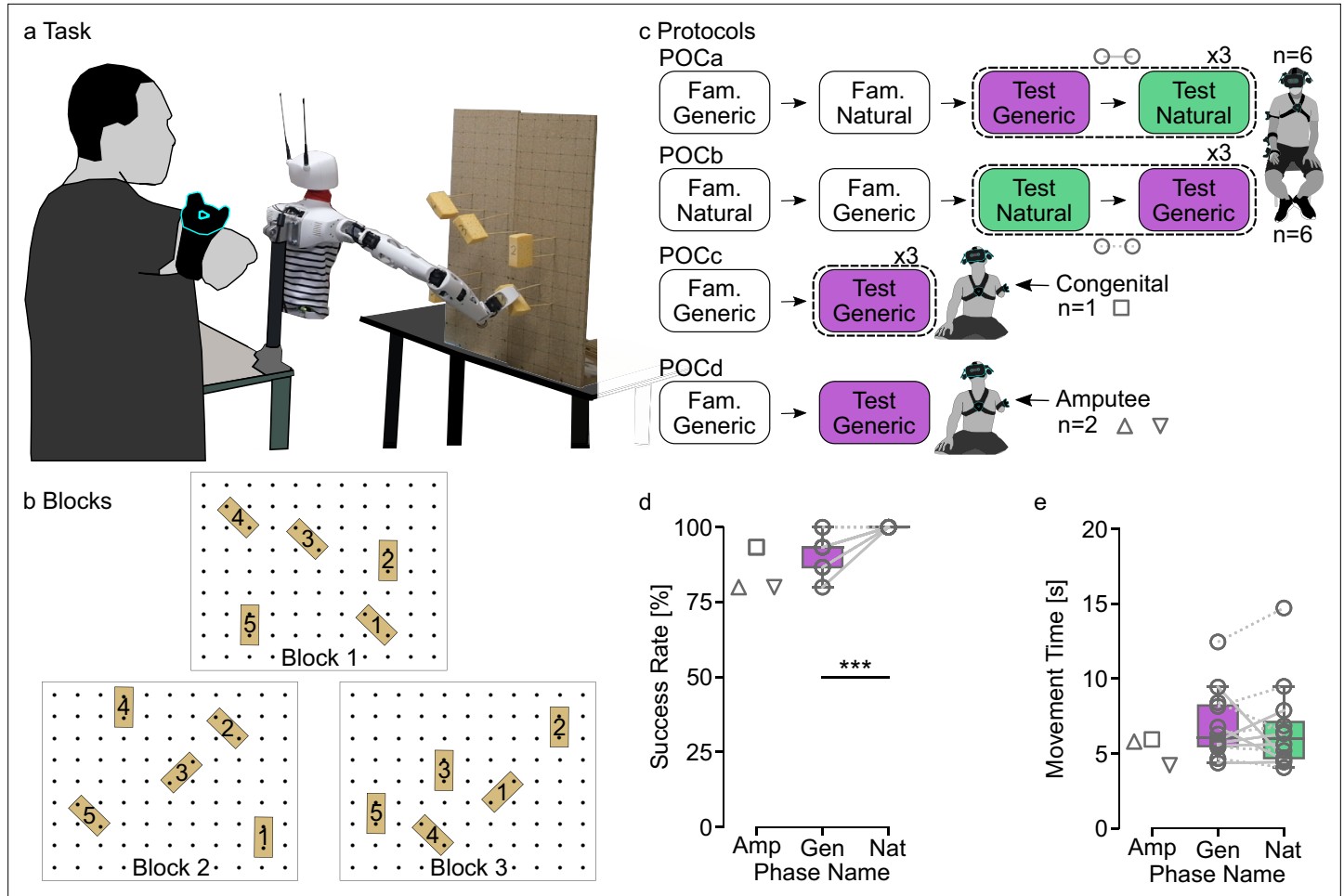

**Figure 4.** Physical Proof of Concept on a tele-operated robotic platform. (**a**) Task and setup. The participant stands still setback from the humanoid robotic platform that faces a board on which five sponges are placed at different positions and orientations. The participant tele-operates the robotic arm so as to reach and grasp each of the five sponges of a block, one trial after another, according to the order indicated by numbers written on sponges. (**b**). Three types of blocks define three spatial arrangements of sponges on the board. (**c**) Protocols of the Proof of Concept (POC) experiments. Each box contains a phase name and the name of the control used, either based on natural arm movements (TestNat) or on predictions from the Generic Artificial Neural Network (ANN) (TestGeneric). Fam. stand for Familiarization phase. The order of test phases was counterbalanced in POCa and POCb. (**d-e**). Results for success rate (**d**) and movement time (**e**). Each gray line corresponds to a participant. In POCa-b, dashed lines indicate participants who began by TestGeneric and plain lines those who began by TestNat. Boxes limits show the first and third quartiles whereas inside line shows the median value. Whiskers show min and max values. Stars represent significant differences, with * for p<0.05, ** for p<0.01, and *** for p<0.001. Triangles represent performances obtained for the Block 1 by the two participants with transhumeral limb loss whereas the square represents performances of the congenital limb different participants on all three Blocks.

## Physical proof of concept on a tele-operated robotic platform

To demonstrate the feasibility of our approach in the physical world, we conducted a Proof of Concept (POC) whereby 15 participants, including two with acquired and one with congenital transhumeral limb difference, were involved in reaching and grasping real objects in various positions and orientations with our novel control applied on a humanoid robotic platform specifically designed to explore human-robot control strategies (*Mick et al., 2019*). As with the Hybrid Arm used to emulate the behavior of an upper arm fitted with a transhumeral prosthesis in virtual reality (*Figure 1e*), the shoulder flexion-extension and abduction-adduction of the robotic arm were operated from real shoulder movement produced by the operator, whereas the five remaining distal joint angles were driven by prediction from the generic ANN (*Figure 4*). In contrast to virtual reality, however, the torso of the humanoid robotic platform was fixed and independent of the operator, such that compensatory movements from the trunk and from shoulder translations were not transmitted to the device. This is important because this implies that the task could only be achieved with the designed control. As in

virtual reality, our novel control solution (TestGeneric) was tested against a control based on natural arm movements (TestNat), whereby all joints of the robotic arm were operated directly from arm movements produced by intact-limbs participants (POC protocols *Figure 4c*).

Despite the functionality of compensatory movement being withdrawn by design, high success rates were achieved by all participants in both conditions (*Figure 4d*). A significant difference was nevertheless found between the proportion of validated targets using our movement-base control and natural control (TestGeneric vs TestNat; medians 93% vs 100%, respectively; n=180; McNemar test, p=0.0003, chi.sq=13.1, df = 1; *Figure 4d*). The congenital limb different participant was able to reach and grasp 93.3% of the targets (i.e. 14/15), and the two participants with transhumeral limb loss were both able to grasp 80% of the first five targets which means that only one target was failed. Furthermore, the validated targets were reached and grasped with movement times similar to those obtained using natural control applied to the robotic arm (POCa-b TestGeneric vs TestNat; medians 6.08 s vs 5.98 s, respectively; two-tailed paired t-test, p=0.7026, t=–0.39198, df = 11; *Figure 4e*). Although no statistical analysis was conducted, the movement times recorded for the congenital limb different participant and for the two participants with transhumeral limb loss were similar to those obtained by the intact-limbs participants with the natural control (Medians: Congenital = 5.95 s; Amp1=5.81 s; Amp2=4.23 s).

## Discussion

Stunning progress has been made in the field of bionic limbs to restore important hand grasping and object manipulation functions, with invasive surgery (*Kuiken, 2009*; *Jönsson et al., 2011*) and implants enabling bidirectional communication with the nervous system (*Ortiz-Catalan et al., 2020*; *Ortiz-Catalan et al., 2014a*; *D'Anna et al., 2019*; *Zollo et al., 2019*). Yet, controlling the numerous joints of a prosthetic arm necessary to place the hand at a correct position and orientation to grasp objects remains challenging, and is essentially unresolved. Here, we provide a non-invasive, movement-based solution to this problem, and a clear demonstration of its effectiveness on individuals with limb differences. Indeed, 29 participants including seven with above-elbow limb loss were able to pick and place bottles in a wide workspace with almost perfect scores (median success rates above 99%) and movement times identical to those of natural movements, while five distal joints of their arm were controlled with our novel solution. The same control principles applied on a robotic platform (*Mick et al., 2019*) also enabled 15 participants, including three with limb differences, to reach and grasp real objects at different positions and orientations, with good success rates and movement times similar to those obtained when the robot was controlled with natural arm movements. This is ahead of other control solutions that have been proposed so far to solve this problem, and whose performances are rarely compared to that of natural movements. We now place this in perspective of recent related works, discussing critical features that enabled such performance, as well as remaining gaps and perspectives for daily-life applications.

### Critical features that unleashed movement-based prosthesis control

The first feature that enabled our movement-based control to work so well compared to previous attempts was to introduce the movement goal as an input to the trained ANN. Without it, the ANN would predict the most likely configuration of the distal joints for a given proximal (shoulder) posture, but irrespective of the location of the object to reach. Although this was found to provide an average distal configuration which was nevertheless suitable to reach in a limited workspace, at the expanse of compensatory movements from the trunk and shoulder (*Mick et al., 2021*), we also showed that adding movement goal as an input greatly improves the performance of this control strategy (*Mick et al., 2021*). Indeed, this enabled to control simultaneously four distal DOFs from elbow to wrist with close to natural coordination (*Mick et al., 2021*), where previous attempts were only able to control one DOF, either the elbow (*Popovic and Popovic, 2001*; *Merad et al., 2020*; *Merad et al., 2018*) or the wrist supination-pronation (*Montagnani et al., 2015b*), or rested upon additional unnatural movements to increase functionality (*Kaliki et al., 2013*). Yet, performance levels were still lower than natural movements, with increased movement times and compensatory movements despite the somewhat limited workspace used, and a control design that was not directly applicable to people with limb loss (*Mick et al., 2021*). Here, we overcame those three limitations.

First, we greatly expanded the applicable workspace of this control. This was achieved by starting from the widest possible workspace, limited by the maximal range of motion of participants, and subsequently using a self-organizing network (*Fritzke, 1994*) to best represent the space covered by participants while producing natural arm movements within this space. Second, we substantially increased the amount of relevant training data, by using the entire trajectories of the recorded natural arm movements, instead of using only arm postures placing the hand sufficiently close to the target (*Mick et al., 2021*). Critically, the use of the entire trajectories was made efficient by artificially placing the target in the hand (see Methods), such that the ANN trained on those data performs a form of natural inverse kinematic solving, i.e., one that provides a solution that is representative of natural arm postures rather than a mere optimization for an arbitrary cost function. Third, we made the control applicable to people with limb loss. Indeed, this was not the case in *Mick et al., 2021* as we used the forearm sensor (not available in people with limb loss) to better assess humeral rotation, which could not be reliably measured from the sole upper arm sensor due to muscles and soft tissues around the humerus. Here, the humeral rotation was transferred as an output of the ANN, being a predicted output rather than a necessary input for the control system. In addition, instead of training the control on natural arm movements specifically produced by the intended user (also not applicable in people with limb loss), we designed a generic control based on natural movements from multiple other individuals, but specifically tuned to the morphology of the user (see Methods). This specific tuning was essential, since a given target position and orientation could call for markedly different arm postures depending on the particular arm morphology of an intended user. Finally, mirror symmetry with respect to the medial plane was applied to accommodate for either side of an amputation. Taken together, all those features enabled our participants, including those with an above-elbow amputation, to reach as well as with their natural arm with our movement-based distal joint prosthesis control.

## Perspectives for daily-life applications

Despite the clear benefits mentioned above, including movement goal as an input of the control system could be seen as a weakness, as it might be difficult to determine in real-life settings. Yet, impressive progress in artificial intelligence and computer vision is such that what would have been difficult to imagine a decade ago appears now well within grasp (*LeCun et al., 2015*). For instance, we showed recently that deep learning combined with gaze information enables identifying an object that is about to be grasped from an egocentric view on glasses (*Pérez de San Roman et al., 2017*), and this even in complex cluttered natural environments (*González-Díaz et al., 2019*). Six-dimensional object pose estimation is also a very active area of computer vision (*Liu, 2022*; *Nguyen et al., 2022*), and prosthesis control strategies based on computer vision combined with gaze and/or myoelectric control for movement intention detection are quickly developing (*Starke et al., 2022*; *Markovic et al., 2015*; *Krausz et al., 2020*; *Ghazaei et al., 2017*; *Mouchoux et al., 2021*; *He et al., 2020*), illustrating the promises of this approach. It remains that generalizing our approach to multiple tasks including more constrained reaches will require future work. For instance, once an intended object has been successfully reached or grasped, what to do with it will still require more than computer vision and gaze information to be efficiently controlled. One approach is to complement the control scheme with subsidiary movements, such as shoulder elevation to bring the hand closer to the body or sternoclavicular protraction to control hand closing (*Kaliki et al., 2013*), or even movement of a different limb (e.g. a foot, *Resnik et al., 2014b*). Another approach is to control the prosthesis with body movements naturally occurring when compensating for an improperly controlled prosthesis configuration (*Legrand et al., 2022*). In both cases, particular attention should be paid to ensure that subsidiary movements do not contaminate natural arm coordination, which is essential to the current movement-based control.

Although our approach enabled participants to converge to the correct position and orientation to grasp simple objects with movement times similar to those of natural movements, it is important to note that further developments are needed to produce natural trajectories compatible with real-world applications. As easily visible in *Videos 2–4*, the distal joints predicted by the ANN are realized instantaneously such that a discontinuity occurs at each target change, whereby the distal part of the arm jumps to the novel prediction associated with the new target location. We circumvented problems associated with this discontinuity on our physical proof of concept by introducing a period before the beginning of each trial for the robotic arm to smoothly reach the first prediction from the ANN. This

issue, however, needs to be better handled for real-life scenarios where a user will perform sequences of movements toward different objects.

Another requirement for our control to be functional on a prosthesis is to have actuated wrist joints available, as those are essential to orient the hand in space (*Montagnani et al., 2015a*; *Kanitz et al., 2018*). This is not the case for most commercial prostheses, which sometimes include wrist flexion-extension (mostly passive), and very rarely wrist radial-ulnar deviation (*Bajaj et al., 2019*). Notable exceptions include the LUKE/DEKA arm (*Resnik et al., 2014a*) and the RIC arm (*Lenzi et al., 2016*), which both include those two degrees of freedom as actuated joints, but with a fixed linear relationship between them. Hopefully, the type of control proposed here will highlight the need for, and foster mechatronic developments of, a suitable actuated wrist with human-like motion capabilities (*Bajaj et al., 2019*; *Fan et al., 2022*).

As already mentioned, the solution proposed here is suitable to control distal arm joints to place the hand at a correct position and orientation to grasp objects in a wide workspace, but not for fine hand and grasp control involved in object manipulations, which relies heavily on tactile and somato-sensory feedback information (*Marasco et al., 2021*; *Marasco et al., 2018*). In this context, our movement-based approach appears complementary to more invasive ones, which specifically target those latter functions through bi-directional interactions with the nervous system for both motor control and sensory processing (*Ortiz-Catalan et al., 2020*; *Ortiz-Catalan et al., 2014a*; *D'Anna et al., 2019*; *Zollo et al., 2019*; *Salminger et al., 2019*). Combining those with osseointegration at humeral level (*Ortiz-Catalan et al., 2020*; *Ortiz-Catalan et al., 2014a*) would be particularly relevant as this would also restore amplitude and control over shoulder movements, which are essential for our control but greatly affected by conventional residual limb fitting harnesses and sockets. Yet, testing with a physical prosthesis will need to ensure that the full desired workspace can be obtained with the weight of the attached device, and if not, a procedure to scale inputs will need to be refined. Finally, our movement-based approach could also be combined with semi-autonomous grasp control to accommodate for multiple grasp functions (*Starke et al., 2022*; *Ghazaei et al., 2017*; *He et al., 2020*).

Besides developments needed for application to a real-life setting, the control proposed here could be used as is in virtual reality for the management of Phantom-Limb Pain (PLP), a painful sensation perceived in the missing limb that often occurs after an amputation. Although the precise mechanisms behind PLP and its proposed treatments are still debated and unresolved (*Di Pino et al., 2021*; *Makin and Flor, 2020*), reduction of pain has been repeatedly reported using mirror therapy, whereby the intact hand is moved while the patient views it through a mirror at the place of his/her missing limb (*Chan et al., 2007*). Yet, mirror therapy was found ineffective on patients with distorted (telescoped) phantom limb (*Foell et al., 2014*), and is not applicable to people with bilateral limb loss. Those two limitations can easily be overcome in virtual reality (*Thøgersen et al., 2020*), and our novel movement-based control provides a solution immediately available to control virtual (missing) limbs with natural coordination solely from residual limb motion.

Importantly, self-reported feedback from amputated participants indicates that overall, they found our prosthesis control solution intuitive and natural, and would use it should it be available on their prosthesis (see Results, Successful validation on individuals with limb loss). Given the rich perspectives associated with this movement-based control alternative, its complementarity with other quickly developing approaches, and the demonstration provided here of its effectiveness on people with limb loss, we believe that this alternative is going to positively impact the field of bionic limbs and prosthesis control.

## Methods

### Participants

All participants had normal or corrected-to-normal vision and none suffered from a motor disorder that could have affected their ability to perform the task (except limb difference in Exp3 and in the physical proof of concept POC). The intact-limbs participants' handedness was assessed using the Edinburgh Handedness Inventory (EHI) (*Oldfield, 1971*). For Exp1 and 2, EHI scores over 50 (below –50) corresponded to right-handed (left-handed) participants. For POCa-b, intact-limbs participants with positive EHI scores were included. Exp1 was conducted on 10 naive, intact-limbs, right-handed

participants (five males, EHI mean 84.0; SD 18.4), aged 24–43 years (mean 27.3; SD 6.0). Exp2 was conducted on 12 naive, intact-limbs participants (eight males), aged 20–35 years (mean 24.1; SD 4.4). Six of them were right-handed (EHI mean 96.7; SD 5.2), and the other six were left-handed (EHI mean –85.4; SD 13.0). Exp3 was conducted on seven naive participants having undergone transhumeral amputation ( seven males), aged 25–48 (mean 40.4; SD 8.4). Information related to each participant's amputation is provided in *Table 1*. POCa-b was conducted on 12 naive, intact-limbs participants (six males, EHI mean 87.1; SD 22.8), aged 19–69 years (mean 33.3; SD 16.6). POCc was conducted on one congenital limb different participant, with forearm malformation on the right side, male, aged 22, naive about the task. POCd was conducted on two male participants who had undergone trans-humeral amputation on the right side, aged 34 and 39 years. Both were also included in the Exp3 (see *Table 1* lines 4 and 5). They completed Exp3 before POCd. All participants gave their informed consent and the research presented here has been conducted in accordance with the Declaration of Helsinki and with a local ethics committee (CCP Est II: n°2019-A02890-57).

## Apparatus

During an experimental session, participants remained seated on a chair located at the center of the experimental room. They wore a virtual reality headset (Vive Pro, HTC Corporation) that was adjusted by the experimenter to fit the head firmly and comfortably. When movements of the dominant arm for intact-limbs participants or the valid arm (contralateral to the amputated side) for people with transhumeral limb loss were recorded, four motion trackers (Vive Tracker HTC Corporation) were attached to the body using elastic straps. Each segment of the arm (upper arm, forearm, and hand) as well as the trunk had a dedicated tracker attached to it. The fingers were immobilized with hand wraps so that the hand tracker would move with wrist movements only. For people with transhumeral limb loss, when the motion of the amputated side was recorded, only two trackers were attached: one on the trunk and one on the residual limb. A push-button was placed under the participant's contralateral hand or under the participant's dominant foot.

The infrared beacons and virtual environment were calibrated so that the workspace was centered on the chair, its ground plane at the same height as the room's floor, and its scale identical to real-world dimensions. For each VR device (headset and trackers), the tracking setup measured the 3D position and orientation relative to a fixed reference frame within the virtual environment, using SteamVR (Valve Corporation) as middleware. These measurements were recorded at 90 Hz and the virtual environment was displayed synchronously to the participant at a 90 Hz refresh rate through the headset's stereoscopic display. The Unity engine (Unity Technologies) was used to run the simulation of the virtual scene's contents and interaction with the participant.

## Virtual arm calibration

The scene displayed a virtual arm whose skeleton consisted of three rigid segments (upper arm, forearm, and hand) linked to each other by spherical joints. After the participant was equipped with the VR devices, a procedure was carried out to make this virtual arm mimic the participant's actual arm motion. This procedure included five steps:

1. During a ten-second recording, motion data were collected while the participant was asked to perform slow movements using all of the arm's degrees of freedom (DoFs): shoulder flexion-extension ($\theta_{S-FE}$), shoulder abduction-adduction ($\theta_{S-AA}$) and humeral rotation ($\theta_{H-R}$), elbow flexion-extension ($\theta_{E-FE}$), forearm pronation-supination ($\theta_{F-PS}$), wrist flexion-extension ($\theta_{W-FE}$) and radial-ulnar deviation ($\theta_{W-RU}$) (see *Figure 1—figure supplement 1a*). For participants with arm amputation, when residual limb movements were recorded, only the first two DoFs were taken into account.

2. The method described in *O'Brien et al., 1999* was applied to estimate the joint centers' locations relative to a parent tracker. The upper arm's tracker worked as the parent for the virtual shoulder and elbow, whereas the forearm's tracker worked as the virtual wrist's parent. At the end of this step, the estimated joint centers were displayed as yellow spheres linked by gray lines and the trackers' silhouettes were outlined in the virtual scene. For people with transhumeral limb loss, when residual limb movements were considered, only the shoulder's center was estimated, with the upper arm's tracker working as a parent. Based on the tracker's orientation, a gray line drawn in the virtual scene indicated the estimated actual arm's humeral axis. The line's length was estimated based on the participant's height (see 3). A yellow sphere representing a

hypothetical elbow center was placed according to these estimations of the humerus's orientation and length as well as the shoulder center.

3. The virtual arm's segment dimensions were adjusted to match those of the participant's arm. These dimensions were measured as the distances between estimated joint centers. When residual limb movements were used on people with transhumeral limb loss, these dimensions were computed by scaling a standard set of segment lengths based on the participant's height.

4. Then, the virtual shoulder was attached to the participant's estimated shoulder center, so that the root of the virtual arm would follow the actual shoulder at all times.

5. The virtual arm was locked in a reference posture with the elbow flexed at 90° where its segment orientations and joint positions were clearly visible. The virtual arm's segments were 'linked' to the corresponding trackers one at a time, while the yellow spheres acted as anatomical landmarks. First, the participant was asked to move their arm so that the yellow sphere representing the estimated elbow center overlaid the virtual arm's elbow. When the overlaying was deemed correct, the virtual upper arm became a child object of the corresponding tracker, so that its orientation followed that of the actual upper arm. Then, the same method was repeated to associate the virtual and actual forearms by overlaying the estimated wrist's yellow sphere with the virtual arm's wrist. Finally, the tracker's silhouette was used as a landmark for the participant to orient the actual hand similarly to the virtual hand, aligned with the virtual forearm. The procedure ended with the virtual hand being made a child object of the hand tracker. For people with transhumeral limb loss, when residual limb movements were considered, only the virtual upper arm needed to be attached to the corresponding tracker. The hypothetical elbow sphere was used as a landmark for participants to align their residual limb with the virtual upper arm.

As a final step, this virtual arm calibration was corrected for errors in humeral rotation, and reduced from a 9-DoF to a 7-DoF virtual arm. Indeed, soft tissues around the biceps and triceps are such that the sensor attached to the upper arm is not able to follow accurately the rotation of the humerus. To counter this, humeral rotation was computed based on the triangle formed by the centers of the three joints (shoulder, elbow, and wrist), estimated using both the upper arm and the forearm sensors. Furthermore, the procedure described above considers three rigid segments linked by spherical joints offering each three DoFs in rotation. Although the resulting nine DoFs allowed the arm's segments to be placed at all times in orientations identical to those of the actual arm despite slight variations from an ideal 7-DoF arm, the reduction to seven DoFs was necessary to match anatomical arm description, and to emulate control over the relevant prosthesis joints. This was achieved by extracting seven joint angles from the 9-DoF kinematic chain's segment orientations, following a kinematic model comprising three DoFs at shoulder level, one DoF at elbow level, and three DoFs at wrist level. Then, the same model was used to compute the segment orientations corresponding to the posture described by these seven joint angles, and the resulting virtual arm was moved accordingly in the virtual scene.

## Hybrid arm control

The Hybrid Arm was designed to emulate the behavior of a residual upper arm fitted with a transhumeral prosthesis (*Figure 1e*). In the case of an actual prosthesis, movements of the whole arm would combine the wearer's residual limb motion with the prosthesis's actuation of artificial joints, hence the term 'Hybrid.' To emulate this behavior, the two most proximal DoFs (i.e. shoulder flexion-extension and shoulder adduction-abduction) were taken from the participant's natural shoulder motion derived from the Virtual Arm, whereas the five remaining joint angles were driven by predictions from an ANN trained as indicated in the next section (*Figure 1c*). Following the same 7-DoF model as with the Virtual Arm, segment orientations were then computed from the whole set of seven joint angles, and the Hybrid Arm was moved accordingly in the virtual scene.

## Own and generic ANN

In order to drive the Hybrid Arm's five distal joints, an ANN was trained to predict the five corresponding joint angles from natural arm movements recorded using the Virtual Arm in the VR setup. This section presents the two ANNs used in this study: The Own ANN, trained on the data produced by the same participant as the one that is going to use the network to control the Hybrid Arm, and the Generic ANN, trained on data from 10 other participants recorded in Exp1, and tuned to the arm size of the user. ANNs inputs and outputs are presented in *Figure 1c* and in *Figure 1—figure supplement 1a*. The network structure includes two densely connected layers of 256 neurons each, a dropout

layer with a drop fraction of 0.5, and a dense layer of 64 neurons. The network was implemented and trained using TensorFlow in association with Keras as the programming interface.

The Own ANN training data was taken from the recording of an Initial Acquisition phase performed with the Virtual Arm (cf. Experimental Phases). From this recording, seven signals were extracted and fed to the ANN as inputs: the two most proximal angles of the Virtual Arm, and five goal-related *contextual information* (three Cartesian coordinates and two spherical angles that define the position and orientation of the hand as if a hypothetical cylindrical target was placed in it at any time, see an explanation for this choice in the next paragraph). The error between the ANN outputs (i.e. predictions of the five distal DoFs) and the actual five distal joints of the Virtual Arm produced in the same recording session were used to train the ANN. The network was trained during a pause after the Initial Acquisition phase, and was, therefore, specific to the corresponding participant.

In our experiments, the targets were sparse and scattered within a wide and continuous workspace. Mirroring this discrete distribution, the goal-related contextual information describing the target locations provided discontinuous and highly clustered signals, displaying little variability within a trial and changing abruptly to express the new target location as soon as the next trial began. Preliminary testing revealed that training on such input signals resulted in the ANN being much more subject to overfitting and less efficient for control. To avoid this issue, the training data made use of hand locations instead of target locations to provide contextual information. For each sample, the recorded arm posture was, therefore, treated as if it brought the virtual hand to the exact location of a hypothetical target. Accordingly, the contextual information provided as input corresponded to the position and orientation of the virtual hand, such that the training data covered the workspace more homogeneously and continuously. Thirty epochs were done with a learning rate of 1e-4, similar to that used in *Mick et al., 2021*.

In the case of a person with transhumeral limb loss, driving the prosthesis's joints with an ANN trained on the wearer's own motion-tracking data would be impractical. To tackle this, we designed a method to create a Generic ANN, by transforming motion tracking data recorded from previous participants into a training dataset adapted to the current participant. Recordings produced in the Initial Acquisition phase of the 10 participants of Exp1 were concatenated into a single dataset. However, this dataset would not be appropriate 'as is' to train a Generic ANN because the relationship between hand locations and postures depends on arm segment dimensions that differ between participants. Indeed, as illustrated *Figure 1—figure supplement 1b*, similar arm postures give rise to different hand locations depending on arm morphology, such that hand positions need to be 'remapped' to the current participant's arm before being fed as training data to a Generic ANN. This was achieved using forward kinematics solving with the 7-DoF model underlying the Virtual Arm calibrated for the current participant. The dataset adapted to the current participant contained the remapped hand locations as well as the original arm angular configurations from Exp1, and was used to train the Generic ANN to perform the same task as the Own ANN: predict five distal DoFs from two proximal joint angles and five spatial parameters expressing the hand location. For the sake of fair comparison, this network's structure (i.e. layer arrangement and number of neurons) was identical to that of the Own ANN. However, the training dataset included data from 10 participants instead of just one, thus containing approximately 10 times more samples than data used to train an Own ANN. Given this, the epoch number was reduced to 10 to minimize computation time, and a momentum parameter was introduced to further prevent overfitting. An offline analysis indicated that a learning rate of 1.59e-7 combined with a momentum of 0.95 constitutes a good compromise for the Generic ANN to perform well both when the target is considered in the hand (as for the training data used) and when moving toward it (as it is mostly the case during online experimental phases).

## Task

All experiments relied on a pick-and-place task with a virtual cylindrical bottle (*Figure 1b*). Participants were asked to perform the task with either the Virtual Arm or the Hybrid Arm, by moving their own arm (*Figure 1b*) or residual limb. Even if they were instructed not to move their trunk and to keep their back against the chair depending on the phase and protocol, participants were not physically restrained. The goal was to reach and grasp the bottle with the virtual hand, and then bring it to another location indicated by a cylindrical platform. A *trial* refers to only one part of this process: either the bottle-picking or the bottle-placing. In either case, participants completed the trial by

pressing the button while the virtual hand was inside a target zone, corresponding to a region in the five-dimensional space of hand locations (3D position × 2D orientation) centered on the target's location and delimited by a spatial and angular tolerance. A hard constraint was defined with a spatial tolerance of 2 cm and an angular tolerance of 5°, whereas a relaxed constraint was defined with a spatial tolerance of 4 cm and an angular tolerance of 10°. A semi-transparent arrow was attached to the virtual hand to indicate the hand's axis (arrow direction) and center (arrow base) to help participants bring the hand inside the target zone. Whenever the hand was inside the target zone, the bottle turned red as a sign that it could be either grasped or released. The virtual hand was limited to two states: either open and empty, or closed and holding the bottle. Participants could only toggle the hand's state while the hand was in the target zone, by pressing the button to complete the trial. During a bottle-picking trial, the target corresponded to the bottle itself: the target's center was placed at the middle of the bottle's height and its axis was the bottle's revolution axis. During a bottle-placing task, the target corresponded to the cylindrical platform on which the bottle needed to be placed: the target's axis was perpendicular to the platform's surface and its center was placed so that a correct hand positioning would bring the bottom of the bottle against the platform. This was made so that the instruction to 'place the bottle on the platform' would remain intuitive.

Participants were given a short time (either 5 or 10 s depending on the experimental phase) to complete each trial, and instructed to perform the task at a comfortable yet sustained pace. If the task was not completed within the allotted time, the current trial ended with a short audio cue, and the hand's state was automatically toggled. Each experiment involved four to five phases, within which trials were grouped in blocks of 50 trials (i.e. twenty-five repetitions of the pick-and-place process) interspersed with short pauses (usually <1 min, occasionally up to a few minutes if needed).

## Targets sets generation

Two sets of targets were generated for each participant: A set of Plausible Targets for the Initial Acquisition phase (*Figure 1a*), based on the range of motion of joint angles, and a set of Possible Targets for the Test phases, based on movements previously produced in the Initial Acquisition (*Figure 1d*). Each target was defined by five spatial parameters: three Cartesian coordinates of its center (in the shoulder referential of the participant), and two spherical coordinates describing its orientation relative to the vertical axis. Note that because both the bottle and platform are cylindrical, their rotation about their revolution axis is irrelevant to the task, such that a pair of spherical coordinates is sufficient to describe their orientation.

## Joint angle ranges of motion

With the VR headset temporarily taken off, participants were asked to perform a few repetitions of an elementary movement for each arm DoF, traveling across its whole range of motion (*Figure 1— figure supplement 1a*). For each movement, the experimenter performed a demonstration that the participants were required to mimic with their own arm, and ranges of motion were estimated from extreme values reached with the Virtual Arm recorded. In addition, the range of motion of the elbow was artificially fixed at 85% of maximal extension in order to avoid postures in which the arm would be too straight. Indeed, the triangle used to compute humeral rotation (formed by the three joint centers, shoulder, elbow, and wrist) would be too small, or even vanishing for a perfectly straight arm. For participants with transhumeral limb loss, only the first two elementary residual limb movements were performed and subject to range-of-motion extraction.

## Plausible targets set for initial acquisition phase

Plausible targets were generated based on the estimated ranges of motion, as well as on restrictions applied to the workspace. Firstly, 7-DoF arm angular configurations were drawn at random within the ranges of motion following a multivariate uniform probability distribution. Then, forward kinematics was used to compute the target location that would be reached by the virtual hand for those postures. The resulting target locations were then filtered according to three criteria:

- The angle between the target's axis and the vertical axis did not exceed 80°, excluding targets pointing downwards or horizontally.
- The distance between the target's center and the participant's frontal plane exceeded a third of the participant's arm length, ensuring that all targets were in front of the participants, and excluding targets too close to their trunk.

- The distance between the target's center and the horizontal plane passing through the participant's shoulder did not exceed two-thirds of the participant's arm length, excluding targets too close to the legs.

The remaining targets spanned a roughly hemispherical region centered on the shoulder (cf Plausible target set *Figure 1a* and *Figure 2a*). The random drawing went on until 300 suitable targets were obtained, which were then shuffled (half of them treated as picking locations and the other half as placing locations) to form a sequence of alternating bottles and platforms.

## Possible targets set for test phases

A second target set was generated to cover the participant's reachable space more accurately. This was achieved by applying an unsupervised learning algorithm called Growing Neural Gas (GNG) (*Fritzke, 1994*) to arm angular configurations previously recorded in the Initial Acquisition phase. A GNG is a type of self-organizing map whose structure is based on a graph where each node is associated with a position in the feature space (in our case, the 7-dimensional space of joint angles describing an arm posture). It is trained through a growing process that fits the graph's topological structure to the input data by incrementally moving existing nodes and adding new ones. This process allows the graph to 'learn' the input data's topology in terms of size and local density, and returns a set of nodes directly within the feature space by the end of the training (*cf* red dots *Figure 1d*).

In the present case, the input data corresponded to the Virtual Arm's postures performed by the participant during the Initial Acquisition phase, downsampled by a factor of 10 for the growing process to remain time-efficient. In this way, the neural gas grew inside the region of the configuration space effectively explored when the participant moved their arm to complete the task. The training parameters were tuned to return 200 nodes, and the 7-DoF postures associated with these nodes were transformed into a set of 200 Possible targets using forward kinematics solving (cf Possible target set *Figure 1d* and *Figure 2a*). The generated targets were then ordered in a sequence by randomly drawing targets from the set in a way that prevented two consecutive targets from being too close (<20 cm) to each other.

Because participants with arm amputation did not perform the Initial Acquisition phase, the GNG was applied to data recorded from previous participants of Exp1, following a similar reasoning as behind the Generic ANN. The data from the Initial Acquisition phases of the 10 participants for Exp1 were then filtered as exposed previously for plausible targets (except that ranges of motion associated with residual limb motion were applied), and downsampled by a factor of 100 to obtain an amount of input samples comparable to that used in Exp1 and 2 (where data from a single recording session was downsampled by a factor of 10). The growing process and target generation from the resulting postures followed the same method as explained previously, except that mirror symmetry with respect to the medial plane was applied as appropriate to accommodate the amputated side and the valid arm side used for participants in Exp3.

## Experimental phases

### Familiarization

The first phase was designed to allow participants to familiarize themselves with the apparatus, virtual scene, and experimental task. During this phase, intact-limbs participants drove the Virtual Arm to perform up to three blocks of 50 trials while target locations followed the first items of the Plausible Targets Set. Hard Constraints (2 cm, 5°) were applied to the target zone to ensure maximal use of the range of motion. The time limit was set at 5 s, with the experimenter being able to manually skip a trial upon request in the event of a participant having issues completing this trial. The Familiarization phase ended when the participant was able to reach most targets comfortably.

For participants with transhumeral limb loss, the 5 s time limit was withdrawn so that they could freely explore the apparatus. Relaxed constraints (4 cm, 10°) were applied to mimic the constraints applied during the following Test phases in Exp3. For the residual limb side, participants drove the Hybrid Arm using predictions from the Generic ANN to reach targets from the possible targets set. For the valid side, the Familiarization phase was done on the same targets after they were mirrored symmetric with respect to the medial plane, and the participant drove the Virtual Arm.

## Initial Acquisition

The aim of the Initial Acquisition phase was to record participants' natural movements in order to train their Own ANN used in Exp1 and 2, and to train the Generic ANN used in Exp2 and 3. The participant controlled the Virtual Arm while the Plausible Targets Set was used to elicit 300 trials. As in the Familiarization phase, hard constraints (2 cm, 5°) were applied with a 5 s time limit. In order to promote arm movements only, participants were asked to keep their backs against the chair and not to move their trunks.

## Test

The Test phases aimed at comparing performances achieved using the Hybrid Arm or the Virtual Arm. When using the Hybrid Arm, either the Own or the Generic ANN was interfaced so that at each time step, the ANN received seven inputs and predicted five joint angles. As in the training data, two of these inputs were the proximal joint angles extracted from the actual shoulder's motion and mimicked by the Hybrid Arm's proximal DoFs. However, the contextual information provided by the five remaining inputs was different from that of the training data. Indeed, instead of expressing the hand location, they expressed the target location (bottle or platform), thereby being congruent with the current goal of the task. The distal joint angles predicted by the ANN were then sent back to the simulation engine in order to update the Hybrid Arm's posture. As in the Initial Acquisition phase, participants were required to perform the task by moving their own arm in order to bring the virtual hand on the target, with the help of the semi-transparent arrow. They were given no details regarding the operation of the Hybrid Arm, and instructed to complete the task by performing arm movements as natural as possible. For test phases of Exp2 and 3, the instruction 'to keep their back against the chair and not to move their trunk' was somehow relaxed such that they were allowed to move their trunk only if deemed absolutely necessary to reach the target. One Test phase consisted of 200 trials corresponding to the Possible Targets Set, conducted with relaxed constraints (4 cm, 10°) and a time limit extended to 10 s.

## Protocols

Exp1 aimed at recording natural arm movements from multiple subjects in order to train the Generic ANN for Exp2 and 3, and to compare performances using either the Virtual Arm or the Own ANN. The push-button was placed under the participant's left hand, and each participant performed a Familiarization phase and an Initial Acquisition phase, followed by two Test phases: one with the control of a Hybrid Arm based on the Own ANN predictions, and one with the Virtual Arm (*Figure 3a*). During all those phases, participants were instructed not to move their trunks in order to perform only the arm movement needed to get the target.

Exp2 aimed at comparing performances with the Generic and Own ANNs, and to validate that a Generic ANN trained on right-handed participants could be used by left-handed participants. The push-button was placed on the ground under the participant's dominant foot, and each participant performed a Familiarization phase and an Initial Acquisition phase, followed by three Test phases: One with the Own ANN, one with the Generic ANN, and a final baseline Test phase using the Virtual Arm. The order of Test phases with the Own and Generic ANNs were counterbalanced among participants, with an equal number of left-handed and right-handed participants in each group (*Figure 3a*).

Exp3 aimed at evaluating the performance achieved by participants with transhumeral limb loss using control from the Generic ANN, and compare them to the performance with their valid arm. The push-button was placed under the participants' dominant foot, and participants performed a Familiarization phase followed by two Test phases with the Generic ANN on their amputated side. Then, after proper calibration with their valid arm (i.e. contralateral to the amputation), participants performed a Familiarization phase followed by a baseline Test phase with the Hybrid Arm (*Figure 3a*).

## Data reduction and metrics

Data were first filtered to remove trials with substantial measurement errors associated with motion capture. Two filters were applied: one for 'freezing' behavior and one for 'jumping' behavior. The 'freezing' filter removed trials where a sensor position (e.g. trunk, arm, fore-arm, and hand sensors during baseline phases and only trunk and arm sensors for test phases) stayed still for at least 0.5 s.

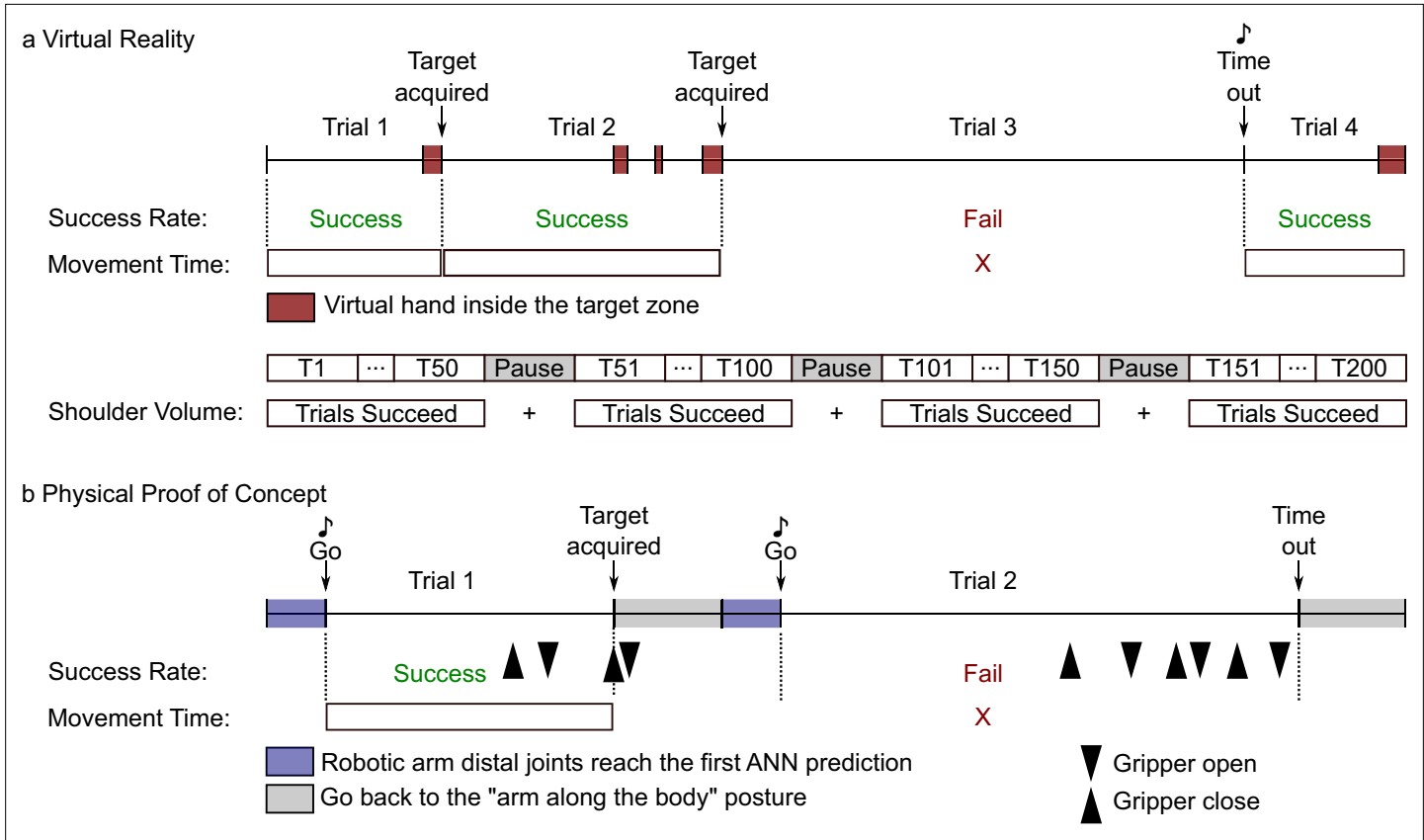

**Figure 5.** Timing protocols of the Virtual Reality (**a**) and the Physical Proof of Concept (**b**) experiments. (**a**) Upper part: Sequence of four hypothetical trials conducted in virtual reality. In each trial, the participant had to move the virtual hand to a target zone. When in the target zone, the cylindrical object turned red (as indicated by the red squares) and the trial was successful if the participant pressed the validation button while within the target zone (see trials 1, 2, and 4). A trial was failed if the participant did not validate the target within the allotted time (see Trial 3). In this case, a sound signaled the time out and the subsequent trial began. Success Rate was calculated for each experimental phase as the percentage of successful trials. Movement Time was computed for each successful trial as the time between the beginning of the trial and the target validation. Lower part: A phase was sliced into blocks of 50 trials. Between blocks, participants could rest during pauses. The Shoulder Volume was computed by pooling all the shoulder movements done during the successful trials of a phase. (**b**). Sequence of two hypothetical trials conducted in the Physical Proof of Concept. Each trial required the participant to move their arm so that a robotic arm could reach a physical target (i.e. rectangular sponges). During the first 0.75 s of a trial, the robotic arm's distal joints reached the first ANN prediction. At this time, a 'Go' signal indicated to the participant that they could start moving. A trial was successful if the participant grasped the target with the robotic gripper and removed it from the wooden sticks (see Trial 1, Target acquired). If the participant was not able to grasp the target within the allotted time, the trial was failed (see Trial 2). Success Rate was calculated for each phase (i.e. a sequence of five targets x three blocks done with the same control type) as the percentage of successful trials. During a trial, the participant was allowed to open and close the gripper as many times as necessary (see black arrowheads). The Movement Time was calculated for each successful trial as the time between the 'Go' signal and the last closure of the gripper. At the end of each trial, the participant was instructed to place their arm alongside their body, while the robotic arm returned to a neutral posture alongside the robotic platform.

The 'jumping' filter removed trials with a shoulder position moving more than 0.01 m between two samples (equivalent to a velocity of 0.9 m/s). Over all experiments, this process removed an average of 3.8 (±7.6) % trials per participant and experimental phase.

Given the high success rate associated with all phases of all experiments (average of 99.22 ± 1.7% trials validated per participant and experimental phase), analyses were conducted on the following metrics computed on trials validated (i.e. button pressed while being within the target zone) in all Test phases by a participant.

The Movement Time (MT) refers to the time taken to reach and validate a target. It is computed as the time between the target appearance and the moment the target is validated using the push-button (see *Figure 5a*).

The shoulder position Spread Volume (SV) is obtained by computing the ellipsoid containing 90% of shoulder position during a phase (see *Mick et al., 2021* for more details and *Figure 5a* for an

illustration of the period over which this was computed). Because a certain amount of shoulder movement naturally occurs during reaching, the SV at baseline should be viewed as a benchmark over which compensatory movements probably occur.

## Physical POC

The feasibility of our approach in the physical world was established through a POC conducted using a humanoid robotic platform with a human-like arm dimension and Degrees of Freedom (*Mick et al., 2019*). The arms of the platform are linked to a fixed robotic trunk with no shoulder translational DoF. Thus, the robotic arm is not worn by the participant, allowing the inclusion of intact-limbs individuals, and preventing the use of compensatory movements from the trunk and shoulder to perform the task. During the experiment, participants stood still, setback to the robotic platform, to avoid visual occlusion of the working space (*Figure 4a*). Only the right arm of the robotic platform was used, and the participants could trigger the opening and the closing of the robotic gripper using a push button placed under their foot. The same virtual reality setup as in Exp1 to 3 was used for the POC (see Apparatus) to link the participant's arm movements to those of the virtual arm. After calibrating the virtual arm (see Virtual Arm Calibration), the headset was removed. Targets were rectangular sponges fixed with wooden sticks on a 45 cm × 40 cm bench (length × height) and a trial was defined as an attempt to reach and grasp a target with the robotic arm. Participants were instructed to place their arms along their bodies at the beginning of each trial and waited for a sound signal (i.e. beginning of the trial) to reach and grasp the target. As in virtual reality, Familiarization and Test phases were conducted to familiarize the participant with the task and to compare controls applied in different Test phases. The phases were divided into blocks of five targets for practical reasons (*Figure 4b*), and numbers written on the sponges indicated the order in which participants had to reach and grasp them within each block of target configurations. The position and orientation of each sponge were set at the beginning of each block using a supplementary sensor. Targets could be vertical or tilted at 45 and –45° on the frontal plane, and varied in depth by about 10 cm. Block 1 was used to familiarize the participants with the control(s) used (*Figure 4c*). In POCa, b, and c, blocks 1–3 were then performed with each control tested, and a Test phase was constituted by pulling trials from all blocks performed with a given control. In POCa-b, participant performed each block of target configuration with both controls before going to the next block of target configuration. Half participants began with the control based on natural arm movements (TestNat) while the other half began with the control based on the Generic ANN (TestGeneric). Participants of POCd only performed the first block for Familiarization and Test phases. The joint configuration of the Virtual Arm (see Hybrid Arm Control) or that of the control based on the Generic ANN (see Own and Generic Arm ANN) was applied to the robotic arm depending on the control tested. The data used to train the Generic ANN were remapped according to the dimensions of the robotic arm. To prevent sharp acceleration at target change when using the control based on the Generic ANN, 0.75 s was allotted before the beginning of each trial (signaled by a sound) for the robotic arm to smoothly reach the first prediction from the ANN. As in Exp1 to 3, only trials validated (i.e. trials with a sponge grasped with the gripper without falling) in all Test phases by a participant were considered for further analysis. MT was defined here as the time spent between the beginning of the trial and the last close of the gripper and was computed for each validated target (*Figure 5b*). Since shoulder translations had no impact on the robotic arm movements, and since the participants' position in the room was not restricted, the shoulder volume SV was not computed.

## Statistical analysis

MT was grouped by participant and test phase, and median values over trials were extracted for each of these groups. By design, the SV already gave a single value per participant and test phase. In this way, we obtained samples of one value per participant for each combination of metric and test phase. For Exp1 and POCa-b, two test phases (with the Virtual Arm and the Own ANN, or with the Virtual Arm and the Generic ANN applied to the robotic arm, respectively) were compared. After testing for normality using the Shapiro test, either a paired t-test or a Wilcoxon test was conducted. For Exp2 and 3, three test phases were compared, involving either the 7-DoF Virtual Arm, the Own ANN, or the Generic ANN. Thus, after testing for normality using the Shapiro Test and for homogeneity of variances between modes using the Maulchly's Test, either a repeated measures ANOVA or a Friedman test was conducted. If a significant difference was found at

this level, post-hoc analyses were conducted using either Tukey tests or Conover tests, respectively. The high success rates observed led to equality between several participants (e.g. at 100% success), which prevented the use of statistical tests based either on normality assumption or on a ranking procedure. Thus, the statistical differences reported here were assessed by comparing the differences in the achievement of each target along all the phases of each experiment. For Exp1 and POCa-b, a McNemar test for paired samples was conducted to find statistical differences between the two phases. For Exp2 and Exp3, a Cochran's Q test for paired samples was first performed followed by a post-hoc McNemar test if needed. Data processing and statistical analysis were carried out with the R software environment, with a significance threshold set at $\alpha=0.05$ with a Bonferroni correction applied if needed. Due to the insufficient number of participants, no statistical analysis was conducted for POCc and d.

## Acknowledgements

The authors would like to thank Émilie Doat and Léa Haefflinger for their help during the experiments, Gerald E Loeb for interactions on an earlier version of the control proposed here and feedback on this manuscript, and all participants who took part in this study. This work was supported by the CNRS interdisciplinary project RoBioVis, and the ANR-DGA-ASTRID grant CoBioPro (ANR-20-ASTR-0012–1).

## Additional information

### Funding

| Funder | Grant reference number | Author |
|---|---|---|
| Direction Générale de l'Armement | ANR-20-ASTR-0012-1 | Vincent Leconte<br>Océane Dubois<br>Rémi Klotz<br>Daniel Cattaert<br>Aymar de Rugy<br>Effie Segas |
| Centre National de la Recherche Scientifique | interdisciplinary project RoBioVis | Sébastien Mick<br>Daniel Cattaert<br>Aymar de Rugy |

The funders had no role in study design, data collection and interpretation, or the decision to submit the work for publication.

### Author contributions

Effie Segas, Conceptualization, Data curation, Software, Formal analysis, Supervision, Investigation, Visualization, Methodology, Writing – original draft, Project administration, Writing – review and editing; Sébastien Mick, Conceptualization, Software, Methodology, Writing – original draft; Vincent Leconte, Resources, Data curation, Software, Formal analysis, Visualization, Methodology; Océane Dubois, Conceptualization, Software, Formal analysis, Investigation, Methodology; Rémi Klotz, Resources, Investigation, Methodology, Project administration; Daniel Cattaert, Conceptualization, Supervision, Project administration, Writing – review and editing; Aymar de Rugy, Conceptualization, Supervision, Funding acquisition, Methodology, Writing – original draft, Project administration, Writing – review and editing

### Author ORCIDs

Effie Segas ⬥ http://orcid.org/0000-0002-1266-0570
Sébastien Mick ⬥ http://orcid.org/0000-0002-9900-6263
Daniel Cattaert ⬥ http://orcid.org/0000-0003-4471-0525
Aymar de Rugy ⬥ http://orcid.org/0000-0001-5645-3680

### Ethics

All participants gave their informed consent and research presented here has been conducted in accordance with the Declaration of Helsinki and with local ethics committee (CCP Est II: n°2019-A02890-57).

Reviewer #1 (Public Review): https://doi.org/10.7554/eLife.87317.3.sa1
Reviewer #2 (Public Review): https://doi.org/10.7554/eLife.87317.3.sa2
Reviewer #3 (Public Review): https://doi.org/10.7554/eLife.87317.3.sa3
Author Response https://doi.org/10.7554/eLife.87317.3.sa4

## Additional files

### Supplementary files
• MDAR checklist

### Data availability
Raw data recorded during the three experiments, and code required for data treatment and ANNs training, are available at https://doi.org/10.5281/zenodo.7187850. Further information and requests should be addressed to the corresponding author Aymar de Rugy (aymar.derugy@u-bordeaux.fr).

The following dataset was generated:

| Author(s) | Year | Dataset title | Dataset URL | Database and Identifier |
|---|---|---|---|---|
| Segas E, Mick S, Leconte V, Klotz R, Cattaert D, de Rugy A | 2022 | Data and code for intuitive movement-based prosthesis control in virtual reality | https://doi.org/10.5281/zenodo.7187850 | Zenodo, 10.5281/zenodo.7187850 |

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
