## [Editor Report · eLife assessment]

This paper's **importance** lies in its integration of movement and contextual information to control a virtual arm for individuals with upper-limb differences. The provided evidence **convincingly** demonstrates the approach's feasibility for manipulating a single object shape in different orientations within a virtual environment. However, additional improvements are needed for this proof-of-concept neuro-model to fulfil practical requirements.

---

## [Referee Report · Reviewer #1 (Public Review)]

Segas et al. present a novel solution to an upper-limb control problem which is often neglected by academia. The problem the authors are trying to solve is how to control the multiple degrees of freedom of the lower arm to enable grasp in people with transhumeral limb loss. The proposed solution is a neural network based approach which uses information from the position of the arm along with contextual information which defines the position and orientation of the target in space. Experimental work is presented, based on virtual simulations and a telerobotic proof of concept.

The strength of this paper is that it proposes a method of control for people with transhumeral limb loss which does not rely upon additional surgical intervention to enable grasping objects in the local environment. A challenge the work faces is that it can be argued that a great many problems in upper limb prosthesis control can be solved given precise knowledge of the object to be grasped, its relative position in 3D space and its orientation. It is difficult to know how directly results obtained in a virtual environment will translate to real world impact. Some of the comparisons made in the paper are to physical systems which attempt to solve the same problem. It is important to note that real world prosthesis control introduces numerous challenges which do not exist in virtual spaces or in teleoperation robotics.

The authors claim that the movement times obtained using their virtual system, and a teleoperation proof of concept demonstration, are comparable to natural movement times. The speed of movements obtained and presented are easier to understand by viewing the supplementary materials prior to reading the paper. The position of the upper arm and a given target are used as input to a classifier, which determines the positions of the lower arm, wrist and the end effector. The state of the virtual shoulder in the pick and place task is quite dynamic and includes humeral rotations which would be challenging to engineer in a real physical prosthesis above the elbow. Another question related to the pick and place task used is whether or not there are cases where both the pick position and the place position can be reached via the same, or very similar, shoulder positions? i.e. with the shoulder flexion-extension and abduction-adduction remaining fixed, can the ANN use the remaining five joint angles to solve the movement problem with little to no participant input, simply based on the new target position? If this was the case, movements times in the virtual space would present a very different distribution to natural movements, while the mean values could be similar. The arguments made in the paper could be supported by including individual participant data showing distributions of movement times and the distances travelled by the end effector where real movements are compared to those made by an ANN.

In the proposed approach users control where the hand is in space via the shoulder. The position of the upper arm and a given target are used as input to a classifier, which determines the positions of the lower arm, wrist and the effector. The supplementary materials suggest the output of the classifier occurs instantaneously, in that from the start of the trial the user can explore the 3D space associated with the shoulder in order to reach the object. When the object is reached a visual indicator appears. In a virtual space this feedback will allow rapid exploration of different end effector positions which may contribute to the movement times presented. In a real world application, movement of a distal end-effector via the shoulder is not to be as graceful and a speed accuracy trade off would be necessary to ensure objects are grasped, rather than knocked or moved.

Another aspect of the movement times presented which is of note, although it is not necessarily incorrect, is that the virtual prosthesis performance is close too perfect. In that, at the start of each trial period, either pick or place, the ANN appears to have already selected the position of the five joints it controls, leaving the user to position the upper arm such that the end effector reaches the target. This type of classification is achievable given a single object type to grasp and a limited number of orientations, however scaling this approach to work robustly in a real world environment will necessitate solving a number of challenges in machine learning and in particular computer vision which are not trivial in nature. On this topic, it is also important to note that, while very elegant, the teleoperation proof of concept of movement based control does not seem to feature a similar range of object distance from the user as the virtual environment. This would have been interesting to see and I look forward to seeing further real world demonstrations in the authors future work.

---

## [Referee Report · Reviewer #2 (Public Review)]

Segas et al motivate their work by indicating that none of the existing myoelectric solution for people with trans-humeral limb difference offer four active degrees of freedom, namely forearm flexion/extension, forearm supination/pronation, wrist flexion/extension, and wrist radial/ulnar deviation. These degrees of freedom are essential for positioning the prosthesis in the correct plan in the space before a grasp can be selected. They offer a controller based on the movement of the stump.

The proposed solution is elegant for what it is trying to achieve in a laboratory setting. Using a simple neural network to estimate the arm position is an interesting approach, despite the limitations/challenges that the approach suffers from, namely, the availability of prosthetic hardware that offers such functionality, information about the target and the noise in estimation if computer vision methods are used. Segas et al indicate these challenges in the manuscript, although they could also briefly discuss how they foresee the method could be expanded to enable a grasp command beyond the proximity between the end-point and the target. Indeed, it would be interesting to see how these methods can be generalise to more than one grasp.

One bit of the results that is missing in the paper is the results during the familiarisation block. If the methods in "intuitive" I would have thought no familiarisation would be needed. Do participants show any sign of motor adaptation during the familiarisation block?

In Supplementary Videos 3 and 4, how would the authors explain the jerky movement of the virtual arm while the stump is stationary? How would be possible to distinguish the relative importance of the target information versus body posture in the estimation of the arm position? This does not seem to be easy/clear to address beyond looking at the weights in the neural network.

I am intrigued by how the Generic ANN model has been trained, i.e. with the use of the forward kinematics to remap the measurement. I would have taught an easier approach would have been to create an Own model with the native arm of the person with the limb loss, as all your participants are unilateral (as per Table 1). Alternatively, one would have assumed that your common model from all participants would just need to be 'recalibrated' to a few examples of the data from people with limb difference, i.e. few shot calibration methods.

---

## [Referee Report · Reviewer #3 (Public Review)]

This work provides a new approach to simultaneously control elbow and wrist degrees of freedom using movement based inputs, and demonstrate performance in a virtual reality environment. The work is also demonstrated using a proof-of-concept physical system. This control algorithm is in contrast to prior approaches which electrophysiological signals, such as EMG, which do have limitations as described by the authors. In this work, the movements of proximal joints (eg shoulder), which generally remain under voluntary control after limb amputation, are used as input to neural networks to predict limb orientation. The results are tested by several participants within a virtual environment, and preliminary demonstrated using a physical device, albeit without it being physically attached to the user.

Strengths:

Overall, the work has several interesting aspects. Perhaps the most interesting aspect of the work is that the approach worked well without requiring user calibration, meaning that users could use pre-trained networks to complete the tasks as requested. This could provide important benefits, and if successfully incorporated into a physical prosthesis allow the user to focus on completing functional tasks immediately. The work was also tested with a reasonable number of subjects, including those with limb-loss. Even with the limitations (see below) the approach could be used to help complete meaningful functional activities of daily living that require semi-consistent movements, such as feeding and grooming.

Weaknesses:

While interesting, the work does have several limitations. In this reviewer's opinion, main limitations are: the number of 'movements' or tasks that would be required to train a controller that generalized across more tasks and limb-postures. The authors did a nice job spanning the workspace, but the unconstrained nature of reaches could make restoring additional activities problematic. This remains to be tested.

The weight of a device attached to a user will impact the shoulder movements that can be reliably generated. Testing with a physical prosthesis will need to ensure that the full desired workspace can be obtained when the limb is attached, and if not, then a procedure to scale inputs will need to be refined.

The reliance on target position is a complicating factor in deploying this technology. It would be interesting to see what performance may be achieved by simply using the input target positions to the controller and exclude the joint angles from the tracking devices (eg train with the target positions as input to the network to predict the desired angles).

Treating the humeral rotation degree of freedom is tricky, but for some subjects, such as those with OI, this would not be as large of an issue. Otherwise, the device would be constructed that allowed this movement.

Overall, this is an interesting preliminary study with some interesting aspects. Care must be taken to systematically evaluate the method to ensure clinical impact.

---

## [Author Response]

The following is the authors’ response to the original reviews.

We are very grateful to the reviewers for their thorough assessment of our study, and their acknowledgment of its strengths and weaknesses. We did our best below to address the weaknesses raised in their public review, and to comply with their recommendations.

**Reviewer #1 (Public Review):**
Segas et al. present a novel solution to an upper-limb control problem which is often neglected by academia. The problem the authors are trying to solve is how to control the multiple degrees of freedom of the lower arm to enable grasp in people with transhumeral limb loss. The proposed solution is a neural network based approach which uses information from the position of the arm along with contextual information which defines the position and orientation of the target in space. Experimental work is presented, based on virtual simulations and a telerobotic proof of conceptThe strength of this paper is that it proposes a method of control for people with transhumeral limb loss which does not rely upon additional surgical intervention to enable grasping objects in the local environment. A challenge the work faces is that it can be argued that a great many problems in upper limb prosthesis control can be solved given precise knowledge of the object to be grasped, its relative position in 3D space and its orientation. It is difficult to know how directly results obtained in a virtual environment will translate to real world impact. Some of the comparisons made in the paper are to physical systems which attempt to solve the same problem. It is important to note that real world prosthesis control introduces numerous challenges which do not exist in virtual spaces or in teleoperation robotics.

We agree that the precise knowledge of the object to grasp is an issue for real world application, and that real world prosthesis control introduces many challenges not addressed in our experiments. Those were initially discussed in a dedicated section of the discussion (‘Perspectives for daily-life applications’), and we have amended this section to integrate comments by reviewers that relate to those issues (cf below).

The authors claim that the movement times obtained using their virtual system, and a teleoperation proof of concept demonstration, are comparable to natural movement times. The speed of movements obtained and presented are easier to understand by viewing the supplementary materials prior to reading the paper. The position of the upper arm and a given target are used as input to a classifier, which determines the positions of the lower arm, wrist and the end effector. The state of the virtual shoulder in the pick and place task is quite dynamic and includes humeral rotations which would be challenging to engineer in a real physical prosthesis above the elbow. Another question related to the pick and place task used is whether or not there are cases where both the pick position and the place position can be reached via the same, or very similar, shoulder positions? i.e. with the shoulder flexion-extension and abduction-adduction remaining fixed, can the ANN use the remaining five joint angles to solve the movement problem with little to no participant input, simply based on the new target position? If this was the case, movements times in the virtual space would present a very different distribution to natural movements, while the mean values could be similar. The arguments made in the paper could be supported by including individual participant data showing distributions of movement times and the distances travelled by the end effector where real movements are compared to those made by an ANN.In the proposed approach users control where the hand is in space via the shoulder. The position of the upper arm and a given target are used as input to a classifier, which determines the positions of the lower arm, wrist and the effector. The supplementary materials suggest the output of the classifier occurs instantaneously, in that from the start of the trial the user can explore the 3D space associated with the shoulder in order to reach the object. When the object is reached a visual indicator appears. In a virtual space this feedback will allow rapid exploration of different end effector positions which may contribute to the movement times presented. In a real world application, movement of a distal end-effector via the shoulder is not to be as graceful and a speed accuracy trade off would be necessary to ensure objects are grasped, rather than knocked or moved.

As correctly noted by the reviewer and easily visible on videos, the distal joints predicted by the ANN are realized instantaneously in the virtual arm avatar, and a discontinuity occurs at each target change whereby the distal part of the arm jumps to the novel prediction associated with the new target location. As also correctly noted by the reviewer, there are indeed some instances where minimal shoulder movements are required to reach a new target, which in practice implies that on those instances, the distal part of the arm avatar jumps instantaneously close to the new target as soon as this target appears. Please note that we originally used median rather than mean movement times per participant precisely to remain unaffected by potential outliers that might come from this or other situations. We nevertheless followed the reviewer’s advice and have now also included individual distributions of movement times for each condition and participant (cf Supplementary Fig. 2 to 4 for individual distributions of movement time for Exp1 to 3, respectively). Visual inspection of those indicates that despite slight differences between participants, no specific pattern emerges, with distributions of movement times that are quite similar between conditions when data from all participants are pooled together.

Movement times analysis indicates therefore that the overall participants’ behavior has not been impacted by the instantaneous jump in the predicted arm positions at each of the target changes. Yet, those jumps indicate that our proposed solution does not satisfactorily reproduce movement trajectory, which has implications for application in the physical world. Although we introduced a 0.75 s period before the beginning of each trial for the robotic arm to smoothly reach the first prediction from the ANN in our POC experiment (cf Methods), this would not be practical for a real-life scenario with a sequence of movements toward different goals. Future developments are therefore needed to better account for movement trajectories. We are now addressing this explicitly in the manuscript, with the following paragraph added in the discussion (section ‘Perspectives of daily-life applications’):

“Although our approach enabled participants to converge to the correct position and orientation to grasp simple objects with movement times similar to those of natural movements, it is important to note that further developments are needed to produce natural trajectories compatible with real-world applications. As easily visible on supplementary videos 2 to 4, the distal joints predicted by the ANN are realized instantaneously such that a discontinuity occurs at each target change, whereby the distal part of the arm jumps to the novel prediction associated with the new target location. We circumvented problems associated with this discontinuity on our physical proof of concept by introducing a period before the beginning of each trial for the robotic arm to smoothly reach the first prediction from the ANN. This issue, however, needs to be better handled for real-life scenarios where a user will perform sequences of movements toward different objects.”

Another aspect of the movement times presented which is of note, although it is not necessarily incorrect, is that the virtual prosthesis performance is close too perfect. In that, at the start of each trial period, either pick or place, the ANN appears to have already selected the position of the five joints it controls, leaving the user to position the upper arm such that the end effector reaches the target. This type of classification is achievable given a single object type to grasp and a limited number of orientations, however scaling this approach to work robustly in a real world environment will necessitate solving a number of challenges in machine learning and in particular computer vision which are not trivial in nature. On this topic, it is also important to note that, while very elegant, the teleoperation proof of concept of movement based control does not seem to feature a similar range of object distance from the user as the virtual environment. This would have been interesting to see and I look forward to seeing further real world demonstrations in the authors future work.

According to this comment, the reviewer has the impression that the ANN had already selected a position of the five joints it controls at the start of each trial, and maintained those fixed while the user operates the upper arm so as to reach the target. Although the jumps at target changes discussed in the previous comment might give this impression, and although this would be the case should we have used an ANN trained with contextual information only, it is important to stress that our control does take shoulder angles as inputs, and produced therefore changes in the predicted distal angles as the shoulder moves.

To substantiate this, we provide in Author response image 1 the range of motion (angular difference at each joint between the beginning and the end of each trial) of the five distal arm angles, regrouped for all angles and trials of Exp1 to 3 (one circle and line per participant, representing the median of all data obtained by that participant in the given experiment and condition, as in Fig. 3 of the manuscript). Please note that those ranges of motion were computed on each trial just after the target changes (i.e., after the jumps) for conditions with prosthesis control, and that the percentage noted on the figure below those conditions correspond to the proportion of the range of motion obtained in the natural movement condition. As can be seen, distal angles were solicited in all prosthesis control conditions by more than half the amount they moved in the condition of natural movements (between 54 and 75% depending on conditions).

**Author response image 1. sa4fig1:** 

With respect to the last part of this comment, we agree that scaling this approach to work robustly in a real world environment will necessitate solving a number of challenges in machine learning and in particular computer vision. We address those in a specific section of the discussion (‘Perspectives for daily-life application’) which has been further amended in response to the reviewers’ comments. As also mentioned earlier and at the occasion of our reply to other reviewers’ comments, we also agree that our physical proof of concept is quite preliminary, and we are looking forward to conduct future work in order to solve some of the issues discussed and get closer to real world demonstrations.

**Reviewer #2 (Public Review):**
Segas et al motivate their work by indicating that none of the existing myoelectric solution for people with transhumeral limb difference offer four active degrees of freedom, namely forearm flexion/extension, forearm supination/pronation, wrist flexion/extension, and wrist radial/ulnar deviation. These degrees of freedom are essential for positioning the prosthesis in the correct plan in the space before a grasp can be selected. They offer a controller based on the movement of the stump.The proposed solution is elegant for what it is trying to achieve in a laboratory setting. Using a simple neural network to estimate the arm position is an interesting approach, despite the limitations/challenges that the approach suffers from, namely, the availability of prosthetic hardware that offers such functionality, information about the target and the noise in estimation if computer vision methods are used. Segas et al indicate these challenges in the manuscript, although they could also briefly discuss how they foresee the method could be expanded to enable a grasp command beyond the proximity between the end-point and the target. Indeed, it would be interesting to see how these methods can be generalise to more than one grasp.

Indeed, we have already indicated those challenges in the manuscript, including the limitation that our control “is suitable to place the hand at a correct position and orientation to grasp objects in a wide workspace, but not for fine hand and grasp control ...” (cf 4th paragraph of the ‘Perspectives for daily-life applications’ section of the discussion). We have nevertheless added the following sentence at the end of this paragraph to stress that our control could be combined with recently documented solutions for multiple grasp functions: “Our movement-based approach could also be combined with semi-autonomous grasp control to accommodate for multiple graspfunctions (Startke et al., 2022; Ghazaei etal., 2017; He et al., 2020).”

One bit of the results that is missing in the paper is the results during the familiarisation block. If the methods in "intuitive" I would have thought no familiarisation would be needed. Do participants show any sign of motor adaptation during the familiarisation block?

Please note that the familiarization block indicated Fig. 3a contains approximately half of the trials of the subsequent initial acquisition block (about 150 trials, which represents about 3 minutes of practice once the task is understood and proficiently executed), and that those were designed to familiarize participants with the VR setup and the task rather than with the prosthesis controls. Indeed, it is important that participants were made familiar with the setup and the task before they started the initial acquisition used to collect their natural movements. In Exp1 and 2, there was therefore no familiarization to the prosthesis controls whatsoever (and thus no possible adaptation associated with it) before participants used them for the very first time in the blocks dedicated to test them. This is slightly different in Exp3, where participants with an amputated arm were first tested on their amputated side with our generic control. Although slight adaptation to the prosthesis control might indeed have occurred during those familiarization trials, this would be difficult in practice to separate from the intended familiarization to the task itself, which was deemed necessary for that experiment as well. In the end, we believe that this had little impact on our data since that experiment produced behavioral results comparable to those of Exp1 and 2, where no familiarization to the prosthesis controls could have occurred.

In Supplementary Videos 3 and 4, how would the authors explain the jerky movement of the virtual arm while the stump is stationary? How would be possible to distinguish the relative importance of the target information versus body posture in the estimation of the arm position? This does not seem to be easy/clear to address beyond looking at the weights in the neural network.

As discussed in our response to Reviewer1 and now explicitly addressed in the manuscript, there is a discontinuity in our control, whereby the distal joints of the arm avatar jumps instantaneously to the new prediction at each target change at the beginning of a trial, before being updated online as a function of ongoing shoulder movements for the rest of that trial. In a sense, this discontinuity directly reflects the influence of the target information in the estimation of the distal arm posture. Yet, as also discussed in our reply to R1, the influence of proximal body posture (i.e., Shoulder movements) is made evident by substantial movements of the predicted distal joints after the initial jumps occurring at each target change. Although those features demonstrate that both target information and proximal body posture were involved in our control, they do not establish their relative importance. While offline computation could be thought to quantify their relative implication in the estimation of the distal arm posture, we believe that further human-in-the-loop experiments with selective manipulation of this implication would be necessary to establish how this might affect the system controllability.

I am intrigued by how the Generic ANN model has been trained, i.e. with the use of the forward kinematics to remap the measurement. I would have taught an easier approach would have been to create an Own model with the native arm of the person with the limb loss, as all your participants are unilateral (as per Table 1). Alternatively, one would have assumed that your common model from all participants would just need to be 'recalibrated' to a few examples of the data from people with limb difference, i.e. few shot calibration methods.

AR: Although we could indeed have created an Own model with the native arm of each participant with a limb loss, the intention was to design a control that would involve minimal to no data acquisition at all, and more importantly, that could also accommodate bilateral limb loss. Indeed, few shot calibration methods would be a good alternative involving minimal data acquisition, but this would not work on participants with bilateral limb loss.

**Reviewer #3 (Public Review):**
This work provides a new approach to simultaneously control elbow and wrist degrees of freedom using movement based inputs, and demonstrate performance in a virtual reality environment. The work is also demonstrated using a proof-of-concept physical system. This control algorithm is in contrast to prior approaches which electrophysiological signals, such as EMG, which do have limitations as described by the authors. In this work, the movements of proximal joints (eg shoulder), which generally remain under voluntary control after limb amputation, are used as input to neural networks to predict limb orientation. The results are tested by several participants within a virtual environment, and preliminary demonstrated using a physical device, albeit without it being physically attached to the user.Strengths:Overall, the work has several interesting aspects. Perhaps the most interesting aspect of the work is that the approach worked well without requiring user calibration, meaning that users could use pre-trained networks to complete the tasks as requested. This could provide important benefits, and if successfully incorporated into a physical prosthesis allow the user to focus on completing functional tasks immediately. The work was also tested with a reasonable number of subjects, including those with limb-loss. Even with the limitations (see below) the approach could be used to help complete meaningful functional activities of daily living that require semi-consistent movements, such as feeding and grooming.Weaknesses:While interesting, the work does have several limitations. In this reviewer's opinion, main limitations are: the number of 'movements' or tasks that would be required to train a controller that generalized across more tasks and limbpostures. The authors did a nice job spanning the workspace, but the unconstrained nature of reaches could make restoring additional activities problematic. This remains to be tested.

We agree and have partly addressed this in the first paragraph of the ‘Perspective for daily life applications’ section of the discussion, where we expand on control options that might complement our approach in order to deal with an object after it has been reached. We have now amended this section to explicitly stress that generalization to multiple tasks including more constrained reaches will require future work: “It remains that generalizing our approach to multiple tasks including more constrained reaches will require future work. For instance, once an intended object has been successfully reached or grasped, what to do with it will still require more than computer vision and gaze information to be efficiently controlled. One approach is to complement the control scheme with subsidiary movements, such as shoulder elevation to bring the hand closer to the body or sternoclavicular protraction to control hand closing26, or even movement of a different limb (e.g., a foot, cf Resnik et al., 2014). Another approach is to control the prosthesis with body movements naturally occurring when compensating for an improperly controlled prosthesis configuration (Legrand et al., 2022).”

The weight of a device attached to a user will impact the shoulder movements that can be reliably generated. Testing with a physical prosthesis will need to ensure that the full desired workspace can be obtained when the limb is attached, and if not, then a procedure to scale inputs will need to be refined.

We agree and have now explicitly included this limitation and perspective to our discussion, by adding a sentence when discussing possible combination with osseointegration: “Combining those with osseointegration at humeral level (Ortiz-Catalan et al., 2020; Ortiz-Catalan et al., 2014) would be particularly relevant as this would also restore amplitude and control over shoulder movements, which are essential for our control but greatly affected with conventional residual limb fitting harness and sockets. Yet, testing with a physical prosthesis will need to ensure that the full desired workspace can be obtained with the weight of the attached device, and if not, a procedure to scale inputs will need to be refined.”

The reliance on target position is a complicating factor in deploying this technology. It would be interesting to see what performance may be achieved by simply using the input target positions to the controller and exclude the joint angles from the tracking devices (eg train with the target positions as input to the network to predict the desired angles).

Indeed, the reliance on precise pose estimation from computer vision is a complicating factor in deploying this technology, despite progress in this area which we now discuss in the first paragraph of the ‘Perspective for daily life applications’ section of the discussion. Although we are unsure what precise configuration of input/output the reviewer has in mind, part of our future work along this line is indeed explicitly dedicated to explore various sets of input/output that could enable coping with availability and reliability issues associated with real-life settings.

Treating the humeral rotation degree of freedom is tricky, but for some subjects, such as those with OI, this would not be as large of an issue. Otherwise, the device would be constructed that allowed this movement.

We partly address this when referring to osseointegration in the discussion: “Combining those with osseointegration at humeral level (Ortiz-Catalan et al., 2020; Ortiz-Catalan et al., 2014) would be particularly relevant as this would also restore amplitude and control over shoulder movements, which are essential for our control but greatly affected with conventional residual limb fitting harness and sockets.” Yet, despite the fact that our approach proved efficient in reconstructing the required humeral angle, it is true that realizing it on a prosthesis without OI is an open issue.

Overall, this is an interesting preliminary study with some interesting aspects. Care must be taken to systematically evaluate the method to ensure clinical impact.
**Reviewer #1 (Recommendations For The Authors):**
Page 2: Sentence beginning: "Here, we unleash this movement-based approach by ...". The approach presented utilises 3D information of object position. Please could the authors clarify whether or not the computer vision references listed are able to provide precise 3D localisation of objects?

While the references initially cited in this sentence do support the view that movement goals could be made available in the context of prosthesis control through computer vision combined with gaze information, it is true that they do not provide the precise position and orientation (I.e., 6d pose estimation) necessary for our movementbased control approach. Six-dimensional object pose estimation is nevertheless a very active area of computer vision that has applications beyond prosthesis control, and we have now added to this sentence two references illustrating recent progress in this research area (cf. Liu et al., 2022; Ngguyen et al., 2022).

Page 6: Sentence beginning: "The volume spread by the shoulder's trajectory ...".Page 7: Sentence beginning: "With respect to the volume spread by the shoulder during the Test phases ...".Page 7: Sentence beginning: "Movement times with our movement-based control were also in the same range as in previous experiments, and were even smaller by the second block of intuitive control ...".On the shoulder volume presented in Figure 3d. My interpretation of the increased shoulder volume in Figure 3D Expt 2 shown in the Generic ANN was that slightly more exploration of the upper arm space was necessary (as related to the point in the public review). Is this what the authors mean by the action not being as intuitive? Does the reduction in movement time between TestGeneric1 and TestGeneric 2 not suggest that some degree of exploration and learning of the solution space is taking place?

Indeed, the slightly increased shoulder volume with the Generic ANN in Exp2 could be interpreted as a sign that slightly more exploration of the upper arm space was necessary. At present, we do not relate this to intuitiveness in the manuscript. And yes, we agree that the reduction in movement time between TestGeneric1 and TestGeneric 2 could suggest some degree of exploration and learning.

Page 7: Sentence beginning: "As we now dispose of an intuitive control ...". I think dispose may be a false friend in this context!

This has been replaced by “As we now have an intuitive control…”.

Page 8: Section beginning "Physical Proof of Concept on a tele-operated robotic platform". I assume this section has been added based on suggestions from a previous review. Although an elegant PoC the task presented in the diagram appears to differ from the virtual task in that all the targets are at a relatively fixed distance from the robot. In respect to the computer vision ML requirements, this does not appear to require precise information about the distance between the user and an object. Please could this be clarified?

Indeed, the Physical Proof of Concept has been added after the original submission in order to comply with requests formulated at the editorial stage for the paper to be sent for review. Although preliminary and suffering from several limitations (amongst which a reduced workspace and number of trials as compared to the VR experiments), this POC is a first step toward realizing this control in the physical world. Please note that as indicated in the methods, the target varied in depth by about 10 cm, and their position and orientation were set with sensors at the beginning of each block instead of being determined from computer vision (cf section ‘Physical Proof of Concept’ in the ‘Methods’: “The position and orientation of each sponge were set at the beginning of each block using a supplementary sensor. Targets could be vertical or tilted at 45 and -45° on the frontal plane, and varied in depth by about 10 cm.”).

Page 10: Sentence beginning: "This is ahead of other control solutions that have been proposed ...". I am not sure what this sentence is supposed to convey and no references are provided. While the methods presented appear to be a viable solution for a group of upper-limb amputees who are often ignored by academic research, I am not sure it is appropriate for the authors to compare the results obtained in VR and via teleoperation to existing physical systems (without references it is difficult to understand what comparison is being made here).

The primary purpose of this sentence is to convey that our approach is ahead of other control solutions proposed so far to solve the particular problem as defined earlier in this paragraph (“Yet, controlling the numerous joints of a prosthetic arm necessary to place the hand at a correct position and orientation to grasp objects remains challenging, and is essentially unresolved”), and as documented to the best we could in the introduction. We believe this to be true and to be the main justification for this publication. The reviewer’s comment is probably directed toward the second part of this sentence, which states that performances of previously proposed control solutions (whether physical or in VR) are rarely compared to that of natural movements, as this comparison would be quite unfavorable to them. We soften that statement by removing the last reference to unfavorable comparison, but maintained it as we believe it is reflecting a reality that is worth mentioning. Please note that after this initial paragraph, and an exposition of the critical features of our control, most of the discussion (about 2/3) is dedicated to limitations and perspectives for daily-life application.

Page 10: Sentence: "Here, we overcame all those limitations." Again, the language here appears to directly compare success in a virtual environment with the current state of the art of physical systems. Although the limitations were realised in a virtual environment and a teleoperation PoC, a physical implementation of the proposed system would depend on advances in machine vision to include movement goal. It could be argued that limitations have been traded, rather immediately overcome.

In this sentence, “all those limitations” refers to all three limitations mentioned in the previous sentences in relation to our previous study which we cited in that sentence (Mick et al., JNER 2021), rather than to limitations of the current state of the art of physical systems. To make this more explicit, we have now changed this sentence to “Here, we overcome those three limitations”.

Page 11: Sentence beginning: "Yet, impressive progresses in artificial intelligence and computer vision ...".Page 11: Sentence beginning: "Prosthesis control strategies based on computer vision ..."The science behind self-driving cars is arguably of comparable computational complexity to the real-world object detection and with concurrent real-time grasp selection. The market for self-driving cars is huge and a great deal of R&D has been funded, yet they are not yet available. The market for advanced upper-limb prosthetics is very small, it is difficult to understand who would deliver this work.

We agree that the market for self-driving cars is much higher than that for advanced upper-limb prosthetics. Yet, as mentioned in our reply to a previous comment, 6D object pose estimation is a very active area of computer vision that has applications far beyond prosthesis control (cf. in robotics and augmented reality). We have added two references reflecting recent progress in this area in the introduction, and have amended the discussion accordingly: “Yet, impressive progress in artificial intelligence and computer vision is such that what would have been difficult to imagine a decade ago appears now well within grasp (Lecun et al., 2015). For instance, we showed recently that deep learning combined with gaze information enables identifying an object that is about to be grasped from an egocentric view on glasses (Pérez de San Roman et al., 2017), and this even in complex cluttered natural environments (Gonzalez-Diaz et al., 2019) Six-dimensional object pose estimation is also a very active area of computer vision (Liu et al., 2022; Nguyen et al., 2022), and prosthesis control strategies based on computer vision combined with gaze and/or myoelectric control for movement intention detection are quickly developing (Starke et al., 2022; Markovic et al., 2015; Krausz et al., 2020; Mouchoux et al., 2021; He et al;, 2020), illustrating the promises of this approach.”

Page 15: Sentence beginning: "From this recording, 7 signals were extracted and fed to the ANN as inputs: ...".Page 15: Sentence beginning: "Accordingly, the contextual information provided as input corresponded to the ...".The two sentences appear to contradict one another and it is difficult to understand what the Own ANN was trained on. If the position and the orientation of the object were not used due to overfitting, why claim that they were used as contextual information? Training on the position and orientation of the hand when solving the problem would not normally be considered contextual information, the hand is not part of the environment or setting, it is part of the user. Please could this section be made a little bit clearer?

The Own ANN was trained using the position and the orientation of a hypothetic target located within the hand at any given time. This approach has been implemented to increase the amount of available data. However, when the ANN is utilized to predict the distal part of the virtual arm, the position and orientation of the current target are provided. We acknowledge that the phrasing could be misleading, so we have added the following clarification to the first sentence: "… (3 Cartesian coordinates and 2 spherical angles that define the position and orientation of the hand as if a hypothetical cylindrical target was placed in it at any time, see an explanation for this choice in the next paragraph)".

Page 16: Sentence beginning: "A trial refers to only one part of this process: either ...". Would be possible to present these values separately?

Although it would be possible to present our results separately for the pick phase and for the place phase, we believe that this would overload the manuscript for little to no gain. Indeed, nothing differentiates those two phases other than the fact that the bottle is on the platform (waiting to be picked) in the pick phase, and in the hand (waiting to be placed) in the place phase. We therefore expect to have very similar results for the pick phase and for the place phase, which we verified as follows on Movement Time: Author response image 2 shows movement time results separated for the pick phase (a) and for the place phase (b), together with the median (red dotted line) obtained when results from both phases are polled together. As illustrated, results are very similar for both phases, and similar to those currently presented in the manuscript with both phases pooled (Fig3C).

**Author response image 2. sa4fig2:** 

Page 19: Sentence beginning "The remaining targets spanned a roughly ...". Figure 2 is a very nice diagram but it could be enhanced with a simple visual representation of this hemispherical region on the vertical and horizontal planes.

We made a few attempts at enhancing this figure as suggested. However, the resulting figures tended to be overloaded and were not conclusive, so we opted to keep the original.

Page 19: Sentence beginning "The Movement Time (MT) ..."Page 19: Sentence beginning "The shoulder position Spread Volume (SV) ..."Would it be possible to include a traditional timing protocol somewhere in the manuscript so that readers can see the periods over which these measures calculated?

We have now included Fig. 5 to illustrate the timing protocol and the periods over which MT and SV were computed.

**Reviewer #2 (Recommendations For The Authors):**
Minor commentsPage 6: "Yet, this control is inapplicable "as is" to amputees, for which recording ..." -> "Yet, this control is inapplicable "as is" to amputees, for WHOM recording ... "

This has been modified as indicated.

Throughout: "amputee" -> "people with limb loss" also "individual with limb deficiency" -> "individual with limb difference"

We have modified throughout as indicated.

It would have been great to see a few videos from the tele-operation as well. Please could you supply these videos?

Although we agree that videos of our Physical Proof of Concept would have been useful, we unfortunately did not collect videos that would be suitable for this purpose during those experimental phases. Please note that this Physical Proof of Concept was not meant to be published originally, but has been added after the original submission in order to comply with requests formulated at the editorial stage for the paper to be sent for review.

**Reviewer #3 (Recommendations For The Authors):**
Consider using the terms: intact-limb rather than able-bodied, residual limb rather than stump, congenital limb different rather than congenital limb deficiency.

We have modified throughout as indicated.